# A transient postnatal quiescent period precedes emergence of mature cortical dynamics

Soledad Domínguez[1†], Liang Ma[1,2†], Han Yu[3†], Gabrielle Pouchelon[4], Christian Mayer[5], George D Spyropoulos[3], Claudia Cea[3], György Buzsáki[6,7], Gordon Fishell[4,8], Dion Khodagholy[3]*, Jennifer N Gelinas[1,2,9]*

[1]Institute for Genomic Medicine, Columbia University Medical Center, New York, United States; [2]Department of Biomedical Engineering, Columbia University, New York, United States; [3]Department of Electrical Engineering, Columbia University, New York, United States; [4]The Stanley Center at the Broad, Cambridge, United States; [5]Max Planck Institute of Neurobiology, Martinsried, Germany; [6]Neuroscience Institute and Department of Neurology New York University Langone Medical Center, New York, United States; [7]Center for Neural Science, New York University, New York, United States; [8]Department of Neurobiology, Harvard Medical School, Boston, United States; [9]Department of Neurology, Columbia University Medical Center, New York, United States

**Abstract** Mature neural networks synchronize and integrate spatiotemporal activity patterns to support cognition. Emergence of these activity patterns and functions is believed to be developmentally regulated, but the postnatal time course for neural networks to perform complex computations remains unknown. We investigate the progression of large-scale synaptic and cellular activity patterns across development using high spatiotemporal resolution in vivo electrophysiology in immature mice. We reveal that mature cortical processes emerge rapidly and simultaneously after a discrete but volatile transition period at the beginning of the second postnatal week of rodent development. The transition is characterized by relative neural quiescence, after which spatially distributed, temporally precise, and internally organized activity occurs. We demonstrate a similar developmental trajectory in humans, suggesting an evolutionarily conserved mechanism that could facilitate a transition in network operation. We hypothesize that this transient quiescent period is a requisite for the subsequent emergence of coordinated cortical networks.

**\*For correspondence:**
dk2955@Columbia.edu (DK);
jng2146@cumc.columbia.edu
(JNG)

[†]These authors contributed equally to this work

**Competing interests:** The authors declare that no competing interests exist.

## Introduction

Multiple cognitive functions emerge rapidly during early development. Neural networks enable precise spatiotemporal coordination of synaptic and cellular activity in mature brain functions (*Fries et al., 1997*; *Maingret et al., 2016*; *Peyrache et al., 2009*; *Khodagholy et al., 2017*). How immature neural networks develop into their mature form needed for the complex computations underlying cognition remains poorly understood. The first organized and predominant pattern of neural activity that appears in cortical circuits across species is a spindle-like oscillation (10–20 Hz) that occurs intermittently on a background of relative neural inactivity. Known as spindle bursts in rodents and delta brushes in humans, this immature network activity is commonly triggered by peripheral stimuli (*An et al., 2014*; *Hanganu et al., 2006*; *Khazipov et al., 2004*). They demarcate cortical columns or pre-columns and have been linked to neuronal survival (*Golbs et al., 2011*), establishing sensory ensembles, and critical period plasticity (*Khazipov et al., 2004*;

**eLife digest** It can take several months, or even years, for the brain of a young animal to develop and refine the complex neural networks which underpin cognitive abilities such as memory, planning, and decision making. While the properties that support these functions have been well-documented, less is known about how they emerge during development.

Domínguez, Ma, Yu et al. therefore set out to determine when exactly these properties began to take shape in mice, using lightweight nets of electrodes to record brain activity in sleeping newborn pups. The nets were designed to avoid disturbing the animals or damaging their fragile brains.

The recordings showed that patterns of brain activity similar to those seen in adults emerged during the first couple of weeks after birth. Just before, however, the brains of the pups went through a brief period of reduced activity: this lull seemed to mark a transition from an immature to a more mature mode of operation. After this pause, neurons in the mouse brains showed coordinated patterns of firing reminiscent of those seen in adults. By monitoring the brains of human babies using scalp sensors, Domínguez, Ma, Yu et al. showed that a similar transition also occurs in infants during their first few months of life, suggesting that brains may mature via a process retained across species.

Overall, the relative lull in activity before transition may mark when neural networks gain mature properties; in the future, it could therefore potentially be used to diagnose and monitor individuals with delayed cognitive development.

*Colonnese et al., 2010*; *Khazipov et al., 2013*; *Winnubst et al., 2015*). Spindle bursts are characteristic of the first postnatal week of rodent development, and delta brushes disappear shortly after term in human neonates (*Torres and Anderson, 1985*), emphasizing their transient role in network maturation. In contrast, mature cortex exhibits perpetual, complex patterns of neural activity that appear and interact across a wide range of frequencies (*Tort et al., 2009*; *Fujisawa and Buzsáki, 2011*; *Bosman et al., 2012*). The organization of this activity creates precise spatiotemporal windows for neural synchronization, enabling plasticity processes and generation of neural sequences (*Masquelier et al., 2009*; *Geisler et al., 2010*; *Foster and Wilson, 2006*). Such network properties facilitate information processing, and resultant activity patterns have been causally linked to cognitive processes from stimulus perception to learning and memory (*Fries et al., 1997*; *Maingret et al., 2016*; *Peyrache et al., 2009*; *Khodagholy et al., 2017*). Understanding when and how these properties develop in the immature brain is critical given that delay or failure to express mature brain activity is a strong risk factor for subsequent impaired cognition (*Khazipov et al., 2004*; *Anderson et al., 1985*; *Seelke and Blumberg, 2008*; *Holmes and Lombroso, 1993*).

We hypothesized that emergence of these advanced neural network properties could be heralded by the disappearance of immature activity patterns. During this developmental epoch, cortical microcircuits are still undergoing dramatic changes in anatomical connectivity (*Khazipov et al., 2004*) and functional connectivity between and within cortical layers is initiated (*Bureau et al., 2004*) in part through an abrupt increase in synaptogenesis in superficial layers (*Naskar et al., 2019*). Robust GABA-mediated fast inhibition also appears (*Daw et al., 2007*; *Goldberg et al., 2011*) as inhibitory network influences of interneurons increase (*Modol et al., 2020*; *Tuncdemir et al., 2016*; *Favuzzi et al., 2019*). These local circuit changes are paralleled with the gradual ingrowth of subcortical neurotransmitters that assist in setting brain state changes. Together, such modifications could prime a shift in network operating mode, setting the stage for re-emergence of a different oscillation that occupies a similar frequency band, thalamocortical sleep spindles. In contrast to spindle bursts, these oscillations are characteristic of the offline state of non-rapid eye movement (NREM) sleep only, and they facilitate memory consolidation (*Maingret et al., 2016*; *Latchoumane et al., 2017*).

To investigate this hypothesis, we examined neural network dynamics across early rodent and human development. We targeted epochs spanning disappearance of immature activity and emergence of mature sleep patterns in each species. For rodents, we developed conformable, minimally invasive, high-resolution electrocorticography arrays coupled with high-density implantable probes and recorded spontaneous in vivo electrophysiological patterns from unanesthetized mice across the first two postnatal weeks. This approach enabled simultaneous recording of large-scale synaptic and

cellular activity without damaging fragile cortical circuits. We found that the transition between immature and mature network dynamics was characterized by unexpectedly decreased coordinated cellular and synaptic activity at the beginning of the second postnatal week. After this timepoint, precise neuronal synchronization and oscillatory coupling in space and time robustly emerged. Analysis of continuous electroencephalography (EEG) recordings from human subjects, 36–69 weeks post-gestation, revealed a similar developmental trajectory characterized by a transient decrease in neural activity prior to onset of spatiotemporal oscillatory coupling. Therefore, a shift from local, loosely correlated, prominently sensory-driven patterns to internally organized, spatially distributed, and temporally precise activity is preceded by a transient quiescent period. These findings suggest that mechanisms to developmentally regulate functional network capacity may be evolutionarily conserved and manifest on the systems level as similar, discrete states.

## Results

To identify and characterize emergence of advanced neural network properties in the developing brain, we acquired high spatiotemporal resolution electrophysiological recordings from somatosensory cortex of unanesthetized mouse pups aged postnatal day (P) 5 to 14 (n = 108 pups), an epoch that spans the transition from immature to mature activity patterns during sleep. We used minimally invasive surface electrocorticography arrays (NeuroGrids, n = 70 pups) to permit a spatially extensive survey of cortex. These customized NeuroGrids (*Khodagholy et al., 2015*) consisted of 119 electrodes regularly spaced on a diagonal square-centered lattice embedded in 4-μm-thick parylene C to conform to the cortical surface (*Figure 1A*). Recordings were made following recovery from surgery to eliminate any influence of anesthesia (*Figure 1—figure supplement 1* and *Figure 1—figure supplement 2*). Mice at these ages enter into a cyclical pattern of sleep and wakefulness as assayed by peripheral indicators, such as muscular tone, movements, and heart rate (*Figure 1—figure supplement 1*). We ensured electrodes used for analysis were located in somatosensory cortex by postmortem histology. Following NeuroGrid recordings, the location of the array was marked on the surface of the brain using a biocompatible fluorescent material (*Rauhala et al., 2020*) and tissue was harvested for immunohistochemistry. vGlut2 staining identified the location of primary somatosensory and visual cortices relative to NeuroGrid placement on flattened axial slices (*Figure 1B*, upper), and the spatially extensive nature of the array compared to the size of the pups' brains further facilitated anatomical targeting. This histology-based electrode grouping corresponded to spatial localization of electrophysiological activity patterns, identifying electrodes recording from somatosensory cortex (*Figure 1B*, lower; *Figure 1—figure supplement 3*). To additionally capture neural spiking patterns across cortical layers, we stereotactically implanted silicon probes (linear electrode arrangement) in somatosensory cortex of a separate cohort (n = 38 pups). Immunohistochemistry of coronal slices from these brains verified probe placement within somatosensory cortex, permitted allocation of recording electrodes to cortical layers (*Figure 1C*, upper), and demonstrated transcortical neural spiking patterns (*Figure 1C*, lower; *Figure 1—figure supplement 4*).

We expected that physiologically significant features of neural network maturation would be conserved across species. Thus, in parallel, we obtained continuous EEG recordings from normal human subjects 36–69 weeks post-gestation (1 day to 7 months after birth), an epoch that similarly spans the transition from immature to mature activity patterns of human sleep and can be mapped to corresponding timepoints of rodent brain development (*Jouvet-Mounier et al., 1969*; *Clancy et al., 2001*; *Milh et al., 2007*). We first identified epochs of artifact-free behavioral immobility consistent with sleep in the pups (*Figure 1—figure supplement 5*) and compared the activity present with human sleep EEG. Sleep in the youngest mouse pups (P5–7) was characterized by irregular, discontinuous bursts of neural activity (*Figure 1D, E*, upper traces). The predominant activity was organized into spindle bursts, oscillations characteristic of developing rodent sensory cortex (*Figure 1D*, left spectrogram). Sleep in the youngest human subjects recorded was also discontinuous, and contained delta brushes, an oscillatory pattern that develops by 26 weeks post-gestation and disappears around 40–44 weeks (*Khazipov and Luhmann, 2006*; *Figure 1E*, left spectrogram). Delta brushes share morphological and frequency characteristics with spindle bursts. Sleep patterns changed profoundly with maturation in both species. More mature mouse pups exhibited nearly continuous patterns during sleep, an absence of spindle bursts, and the appearance of sleep spindles, which are well-characterized oscillations of NREM sleep (*Figure 1D*, lower trace, right spectrogram). Human

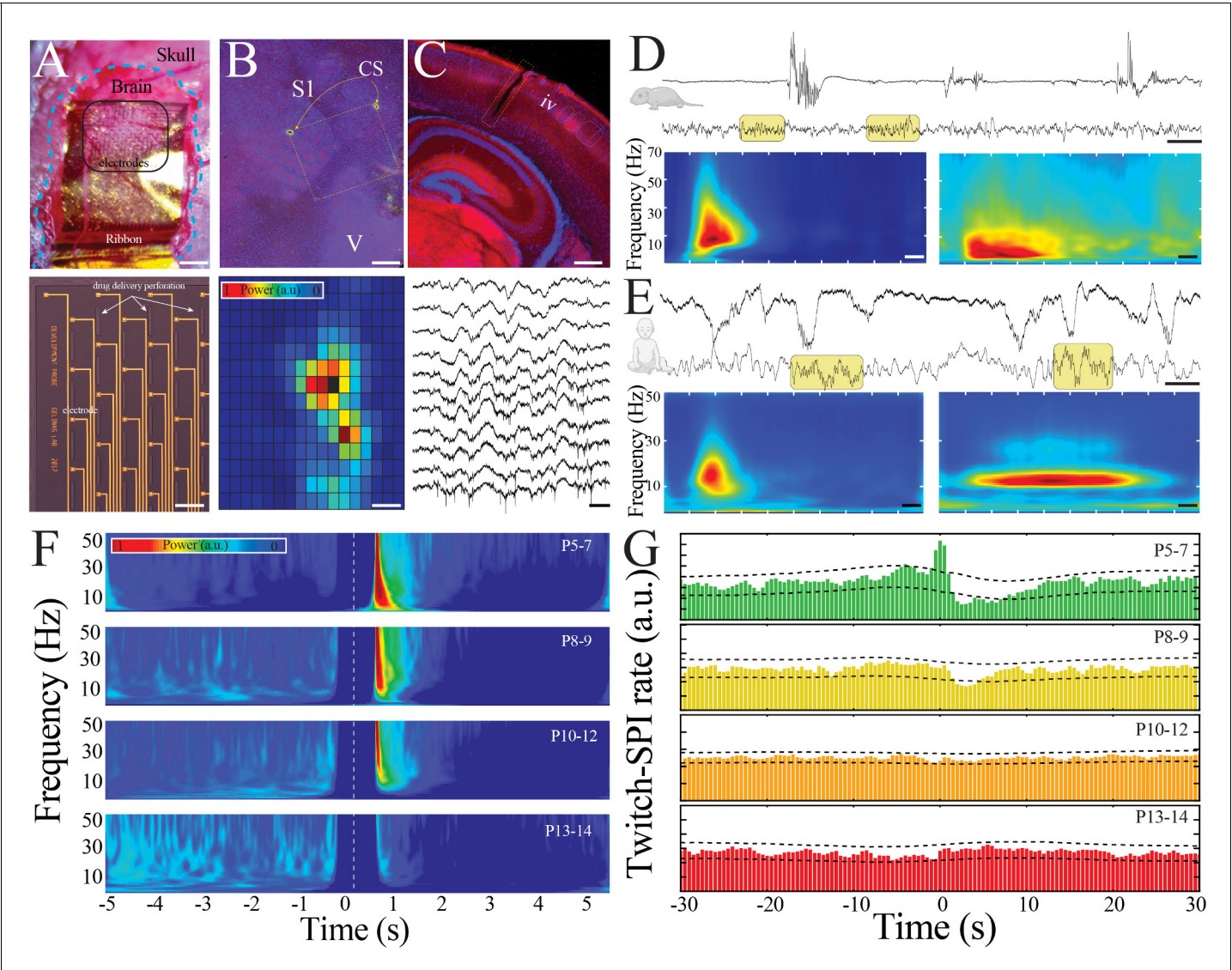

**Figure 1.** Large-scale, high spatiotemporal resolution recording of neural activity across development in rodents demonstrates conserved features with human electroencephalography (EEG). (**A**) Optical micrograph of NeuroGrid conforming to the surface of a P7 mouse pup (upper; scale bar 200 µm). Optical micrograph revealing arrangement of electrodes and perforations in section of NeuroGrid (lower; scale bar 50 µm). (**B**) Fluorescence microscopy of flattened brain slice from P14 mouse pup demonstrating location of NeuroGrid during recording (yellow dashed lines). vGlut2 immunohistochemistry enables identification of visual and somatosensory cortices. S1: primary somatosensory; V: visual; CS: chitosan marking corners of NeuroGrid (upper; scale bar 200 µm). Spatial distribution of spindle band power across a NeuroGrid array in a P7 mouse pup. Warmer colors signify higher power (lower; scale bar 500 µm). (**C**) Fluorescence microscopy of coronal brain slice from P7 mouse pup demonstrating location of implantable silicon probe (yellow dashed lines). vGlut2 immunohistochemistry enables identification of barrels in granular layer of cortex (IV) (upper; scale bar 200 µm). Raw traces across a sample of silicon probe electrodes demonstrating localized neural spiking activity in a P13 mouse pup (lower; scale bar 200 ms). (**D**) Sample raw traces from P5 (upper) and P14 (lower) mouse pups showing shift from discontinuous to continuous activity. Yellow boxes highlight sleep spindles; scale bar 1 s. Spectrograms trigger-averaged on spindle band oscillations (SPIs) in P5 (left) and P14 (right) mouse pups (n = 50 oscillations from each sample session; scale bar 200 ms). (**E**) Sample raw traces from 37-week post-gestational age (upper) and 58-week post-gestational age (lower) human subjects showing shift from discontinuous to continuous activity. Yellow boxes highlight sleep spindles; scale bar 1 s. Spectrograms trigger-averaged on SPIs in 37-week post-gestational age (left) and 58-week post-gestational age (right) subjects (n = 50 oscillations from each sample session; scale bar 200 ms). (**F**) Twitch-triggered spectrograms (twitch center at time = 0, white dashed line) show a progressive decrease in movement-evoked oscillatory activity across development (n = 6716 twitches from n = 52 pups). Note power temporally coincident with twitch is set to zero to avoid any potential artifactual contamination. (**G**) Cross-correlation of twitch and detected SPIs decreases across development. Dashed lines are 95% confidence intervals; time 0 = twitch (n = 6716 twitches from n = 52 pups).

The online version of this article includes the following figure supplement(s) for figure 1:

**Figure supplement 1.** Neurophysiological recording in mouse pups.

*Figure 1 continued on next page*

*Figure 1 continued*

**Figure supplement 2.** Recovery of spindle band oscillation occurrence rate and neural spiking after anesthesia for neural interface device placement.
**Figure supplement 3.** Histological processing and anatomical localization of NeuroGrid electrodes.
**Figure supplement 4.** Localized neural spiking across cortical layers.
**Figure supplement 5.** Isolation of putative quiet/non-rapid eye movement (NREM) epochs in mouse pups.

sleep was similarly continuous, and sleep spindles appear during NREM between 1 and 2 months after term in infants (*Figure 1E*, lower trace, right spectrogram). This mature sleep in rodents and humans is predominantly governed by internally generated dynamics (*Levenstein et al., 2019*). In contrast, self-generated or evoked movements during sleep are closely associated with local field potential (LFP) activity in somatosensory cortex of immature rodents and humans (*Khazipov et al., 2004*; *Dooley et al., 2020*). We confirmed that the association between muscular twitches and changes in LFP power (*Figure 1F*; n = 6716 twitches from n = 52 pups, sampled randomly to ensure uniform group size; Kruskal–Wallis ANOVA, chi-square = 12.93, p=0.0048) or detectable spindle band oscillations (*Figure 1G*; cross-correlation of twitches and spindle band oscillations exceeded 95% confidence interval, black dashed lines, only in pups aged P5–7) decreased as the pups matured. These marked similarities in the electrophysiological features of neural activity during sleep across development suggest the possibility of evolutionarily conserved mechanisms of neural network maturation.

To investigate how this maturation occurs, we examined the macrostructure of sleep LFP with fine temporal precision over early pup development (P5–14). Mature sleep can be classified into distinct stages (NREM and REM) based on electrophysiological criteria, but these indicators do not reliably differentiate the immature analogues of these states (quiet and active sleep, respectively) until after P11 (*Seelke and Blumberg, 2008*). Therefore, we focused on periods of behavioral and EMG quiescence that lasted for longer than 10 s (*Figure 1—figure supplement 5*), with the goal of preferentially analyzing the transition of quiet sleep (which contains spindle bursts) into NREM sleep (which contains sleep spindles). Unexpectedly, recordings across development revealed an epoch of relative neural quiescence at the beginning of the second postnatal week (*Figure 2A*) that corresponded to decreased spectral power and absence of prominent activity at physiological frequencies (*Figure 2B*, *Figure 2—figure supplement 1*). We hypothesized that this quiescence could be related to a transient decrease in the continuity of the neural signal or the power of the activity that was present. Continuity of neural activity was quantified by determining the duration of neural activity above the wideband noise floor. Although continuity increased markedly from the youngest to most mature animals, we found a decrease in the duration of continuous neural activity that corresponded to the identified period of spectral quiescence (*Figure 2C*). When we extracted the average wideband power of the continuous activity that was present at each developmental day, there was also a transient nadir at this timepoint (*Figure 2D*). Because we aimed to characterize the trajectory of the electrophysiological features over time without making a priori assumptions about the nature of the temporal relationship or the rate of change, we employed a modeling approach that considered age as a continuous variable. This approach avoided arbitrary grouping of data from different aged pups without sacrificing statistical power and enabled characterization of the developmental profile beyond testing at individual timepoints. Data were fit with linear and polynomial regression models, and model performance was evaluated using leave-one-out cross-validation (LOOCV) to avoid overfitting (*Figure 2—figure supplement 2*). Polynomial regression provided the best fit for continuity and power data as quantified by the mean squared error (MSE; *Figure 2—figure supplement 2B*). In addition, linear regression was insufficient to model these parameters because linear fit residuals systematically deviated from zero and consistently overestimated observed values at the beginning of the second postnatal week (*Figure 2—figure supplement 2C*). The best fit models exhibited local minima, and to estimate the age at which these nadirs occurred, we employed a bootstrapping method. Bootstrapping data within each developmental day localized probability of nadir in continuity and power at the beginning of the second postnatal week with likelihood far exceeding the distribution obtained by bootstrapping across days (*Figure 2C, D* inset, *Figure 2—figure supplement 2D–F*, *Figure 2—figure supplement 3*). We tested the possibility that the quiescence could be related to a shift in proportion of time spent in each sleep state, a decrease in self-generated sleep

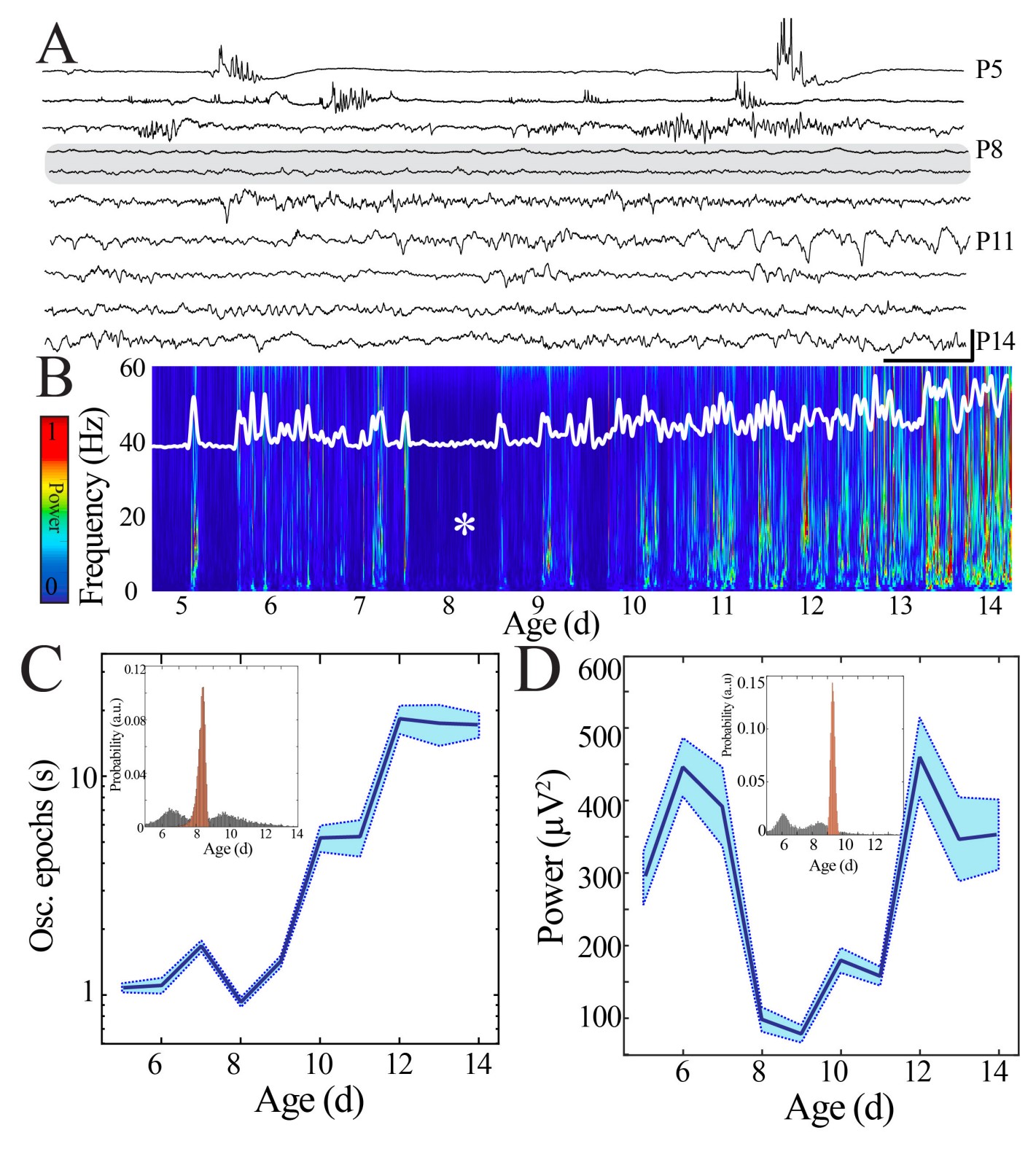

**Figure 2.** Duration and power of cortical oscillatory activity transiently decreases at the beginning of the second postnatal week. (**A**) Sample raw NeuroGrid traces from P5–14 mouse pups demonstrating changing characteristics of oscillatory patterns across development with relative paucity of activity at P8 and P9 (shaded gray box). Scale bar 1 s, 250 µV. (**B**) Compilation of individual spectrograms (60 s duration) from a sample session per day of age demonstrates a transient reduction in local field potential power at the beginning of the second postnatal week. The white superimposed trace represents the overall instantaneous power of neural activity. (**C**) Duration of oscillatory epochs increases nonlinearly across development (blue line =

*Figure 2 continued on next page*

*Figure 2 continued*

mean; shaded blue areas ± SE) with a local minimum at the beginning of the second postnatal week (n = 70 pups). Inset shows probability of a local minimum located at each age for datapoints resampled with replacement across days (gray) compared to datapoints resampled with replacement within days (orange), confirming existence of local minimum between P8 and 9. (**D**) Wideband (WB) power changes nonlinearly across development (blue line = mean; shaded blue areas ± SE), with a local minimum at the beginning of the second postnatal week (n = 70 pups). Inset shows probability of a local minimum located at each age for datapoints resampled with replacement across days (gray) compared to datapoints resampled with replacement within days (orange), confirming existence of local minimum between P8 and 9.

The online version of this article includes the following figure supplement(s) for figure 2:

**Figure supplement 1.** Average power spectra across development.
**Figure supplement 2.** Model fitting and selection with bootstrapping to identify local minima.
**Figure supplement 3.** Model selection and verification of nonlinear developmental trajectory for wideband power in mice.
**Figure supplement 4.** Quantification of sleep proportion and twitch rate in mouse pups.

movements, or differential recovery from anesthesia across ages. In keeping with previous observations, the proportion of active relative to quiet sleep decreased gradually with maturation (***Figure 2—figure supplement 4A***) and we also found a trend toward fewer self-generated sleep movements in the more mature animals, though this shift was not statistically significant (***Figure 2—figure supplement 4B***). Recovery from anesthesia was determined individually for each pup, and there was no difference in post-anesthesia stability of oscillation occurrence rate or neural spiking across pups (***Figure 1—figure supplement 2***). The absence of a marked shift in any of these features during the observed quiescent period makes them unlikely drivers of the phenomenon. Therefore, sleep LFP macrostructure expresses a nonlinear pattern of development that is characterized by a discrete period of relative neural quiescence occurring at the beginning of the second postnatal week.

Rate and synchrony of neural spiking strongly contribute to expression of LFP patterns. We hypothesized that changes in neural spiking patterns could accompany the transient quiescent period. Spiking activity was detected in recordings obtained using implantable silicon probes from two non-contiguous sites (>200 µm vertical separation) within the deep layers of cortex (IV–VI) and one site within the superficial layers of cortex (I–III) per pup recording (n = 38 pups). We first calculated rates of neural spiking only during epochs of above-threshold LFP activity to eliminate effects related to LFP discontinuity. We found that neural spiking increased nonlinearly over the course of development, again with a significant nadir at the beginning of the second postnatal week as quantified using a bootstrapping method (***Figure 3A***, ***Figure 3—figure supplement 1***). These data suggest that lowered intracortical synaptic drive could contribute to the nadir in LFP patterns at the beginning of the second postnatal week. Consistent spiking within neuronal integration time is critical for plasticity processes (***Mainen and Sejnowski, 1995***). To estimate this property, we calculated the inter-spike interval (ISI) between each unique action potential. ISI duration decreased nonlinearly across development, with the longest ISI duration occurring in pups at the beginning of the second postnatal week (***Figure 3B***, ***Figure 3—figure supplement 2***). Population neural spiking at fine time scales is thus both transiently decreased and desynchronized at this time in both superficial and deep cortical layers, resulting in a brief period of relative quiescence prior to the progressive increase in spiking rate and synchrony that accompany subsequent maturation. During mature NREM sleep, there is alternation between periods of neural spiking and hyperpolarization that serves to synchronize neural populations and is implicated in memory consolidation (***Ji and Wilson, 2007***; ***Steriade et al., 1993***). We examined population spiking dynamics in the developing cortex by computing the autocorrelation of spiking activity, which permits quantification of neural synchrony over durations relevant to this slow oscillation (***Figure 3C***). Significant short time scale (<50 ms) interactions (above the 95% confidence interval) robustly emerged only after P9 and were prominent in the most mature pups (***Figure 3D***, left; n = 38 pups, rank-sum z = −4.2, p=2.5×10$^{-5}$). All age groups demonstrated significant suppression of spiking activity within the first 500 ms, becoming more marked as development progressed (***Figure 3D***, right; n = 38 pups, rank-sum z = 3.0, p=0.002). These data implicate that the transient quiescent period demarcates a transition in population spiking that initiates development of a cycle of longer scale network synchronization characteristic of the dynamics observed during mature sleep (***Steriade et al., 1993***).

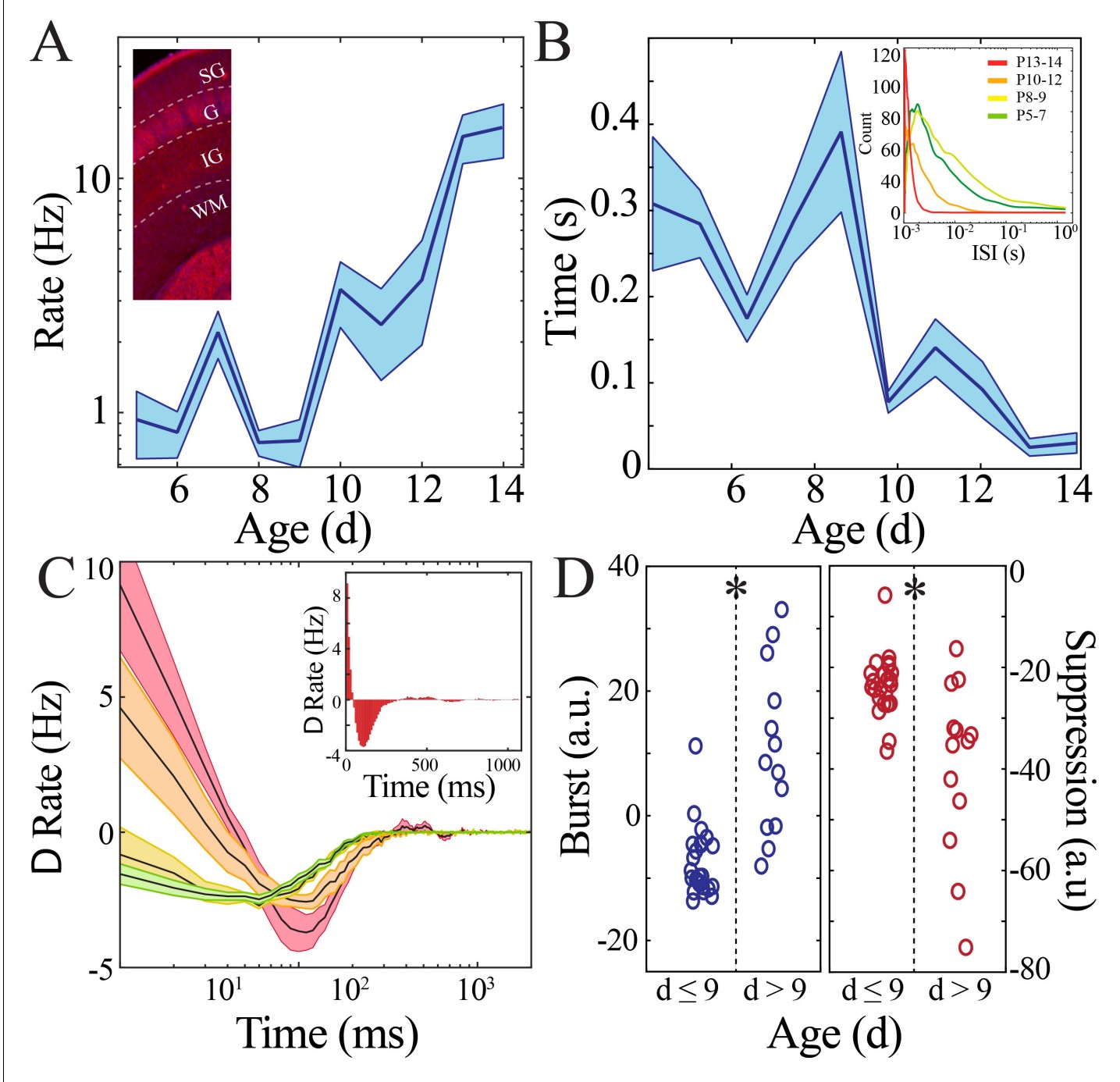

**Figure 3.** Rate and temporal precision of neural spiking increase after the beginning of the second postnatal week. (**A**) Neural spiking rate (layers IV–VI) increases nonlinearly across development (blue line = mean; shaded blue areas ± SE) with a local minimum at the beginning of the second postnatal week (n = 38 pups). Inset shows that vGlut2 immunohistochemistry facilitates identification of layers in barrel cortex (left; SG: supragranular; G: granular; IG: infragranular; WM: white matter). (**B**) Inter-spike interval of spiking activity decreases nonlinearly across development (blue line = mean; shaded blue areas ± SE) with a local maximum at the beginning of the second postnatal week (n = 38 pups). Inset shows histogram distribution of 10,000 inter-spike intervals randomly selected per age group (red = P13–14, orange = P10–12, yellow = P8–9, green = P5–7). (**C**) Average amount of significantly positively correlated (>95% upper confidence interval, corresponding to a rate change > 0) or negatively correlated (<95% lower confidence interval, corresponding to a rate change < 0) neural spiking as computed using spike autocorrelation (traces are mean ± SE). Sample autocorrelation of population neural spiking used for deriving the significant correlations in a P14 pup (inset). Significantly correlated neural spiking at fine-time scales (<50 ms) emerged starting at P10 (n = 38 pups; P5–7 = green, P8–9 = yellow; P10–12 = orange; P13–14 = red). (**D**) Average amount of significant

*Figure 3 continued on next page*

*Figure 3 continued*

positive and negative correlation (outside of 95% confidence intervals) that occurs within 50 ms (burst window; n = 38 pups, z = −4.2, p=2.5e$^{-5}$) and 50–500 ms (suppression window; n = 38 pups, z = 3.0, p=0.002) as computed using spike autocorrelation for immature and mature pups.

The online version of this article includes the following figure supplement(s) for figure 3:

**Figure supplement 1.** Bootstrapping, model selection and verification of nonlinear developmental trajectory for spiking recruitment to neural spiking rate in mice.

**Figure supplement 2.** Bootstrapping, model selection and verification of nonlinear developmental trajectory for spiking recruitment to inter-spike interval in mice.

We then sought to understand the functional relevance of this quiescent transition period to network operations. Key LFP properties (spatial extent, waveform, and power) can provide an index of the underlying neural population activity and its computational purpose – the distance over which neurons can be concurrently recruited, as well as the strength and temporal patterning of this activity (*Lowet et al., 2015*; *Strüber et al., 2017*; *Cole and Voytek, 2017*; *Stark et al., 2014*). We detected spindle band oscillations, which were present in pups of all ages, to facilitate comparisons. We quantified spatial extent using the two-dimensional organization of the NeuroGrid to determine the proportion of electrodes expressing co-occurring spindle band oscillations with a reference electrode located in primary somatosensory cortex (*Figure 4A*; *Dahal et al., 2019*). Waveform was characterized using an asymmetry index (*Figure 4B*) and power was extracted from the filtered Hilbert envelope (*Figure 4C*). Spindle band oscillations in the most immature pups were high power and had a substantially asymmetric waveform, but were expressed over a restricted cortical area (~480 µm diameter). In contrast, spindle band oscillations in the most mature pups were lower power and more symmetric, but could variably be detected over a larger spatial extent (~1.4 mm diameter). Given these observed differences in oscillatory parameters, we asked to what extent they could be used to classify the developmental stage at which the oscillations occurred. A coarse tree model based on spatial extent and waveform asymmetry of manually identified oscillations was effectively trained to classify individual spindle band oscillations as originating from immature (P5–7) or mature (P10–14) pups with 92.4% accuracy (*Figure 4D*), suggesting that these parameters capture defining features of the oscillations at the ends of the developmental spectrum. Spindle band oscillations that occurred during the transient quiescent period at the beginning of the second postnatal week were not consistently classified into either group, instead displaying a unique combination of oscillatory parameters: large spatial extent, low power, and intermediate waveform asymmetry (*Figure 4A–C*, yellow bars). When examined in a combinatorial fashion, a significant inflection point in oscillation properties emerged between P8 and 9 (*Figure 4E*). Polynomial regression provided the best fit for these data (*Figure 4—figure supplement 1*). Bootstrapping data within each developmental day confirmed localization of a nadir at the beginning of the second postnatal week with likelihood far exceeding the distribution obtained by bootstrapping across days (*Figure 4F*). These data indicate that oscillations are transformed at the beginning of the second postnatal week, potentially reflecting a transition in neural network function from consistently highly localized cortical processing to capacity for broader, tuned recruitment of neural activity.

Oscillations in the adult brain precisely organize neural activity across spatial and temporal scales. A key marker for this organization is cross-frequency coupling, which is important for regulating strength of inputs from different afferent pathways (*Schomburg et al., 2014*) and directing flow of information processing (*Buschman and Miller, 2007*). The marked shift in oscillatory properties that we observed at the beginning of the second postnatal week suggested that this organizational capacity may also be developmentally regulated. We first investigated the relationship of spindle band activity with other frequencies by detecting epochs of sustained increase in spindle band power and calculating measures of amplitude cross-frequency coupling. Spindle band epochs in the most immature animals (P5–7) demonstrated high comodulation across a broad, non-specific range of frequencies, in contrast to frequency-specific comodulation expressed in mature animals (P13–14; *Figure 5A*). Specific, discrete frequency bands that demonstrated significant temporal coupling with spindle band oscillations emerged only after P8–9. Discrete coupling with the gamma band (45–60 Hz) first occurred at P10–12, with an additional high gamma band (60–90 Hz) arising at P13–14 (*Figure 5B*; n = 1020 spindle band oscillations from 51 pups; area under the curve [AUC] for 45–60 Hz chi-square = 10.13, p=0.02; AUC for 60–90 Hz chi-square = 24.29, p<0.001). Taken together,

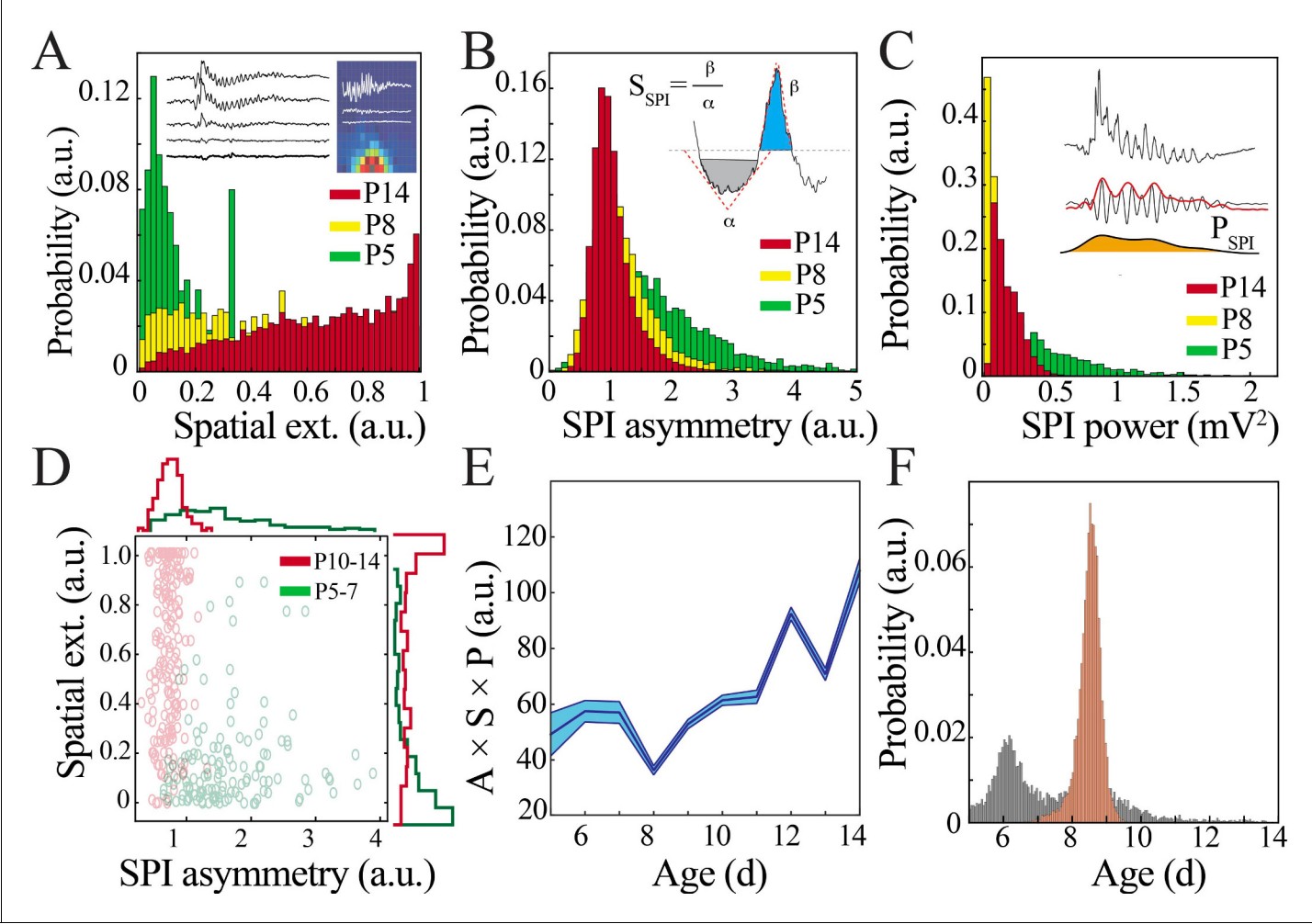

**Figure 4.** Key functional oscillatory properties transition at the beginning of the second postnatal week. (A) Magnitude and variability of spindle band oscillation spatial extent varies across pup development (n = 44 pups, 42,335 spindle band oscillations; Kolmogorov–Smirnov [KS] statistic P5 vs. P8 = 0.47, p<0.001; P5 vs. P14 = 0.71, p<0.001; P8 vs. P14 = 0.26, p<0.001; all p-values adjusted for multiple comparisons using Bonferroni correction). Inset shows sample raw traces from spatially distributed NeuroGrid electrodes (left) and average spindle band power across NeuroGrid array (each square represents one NeuroGrid electrode; warmer traces indicate higher power) from a P5 pup. (B) Waveform asymmetry of spindle band oscillations decreases across pup development (n = as in *Figure 3A*; KS statistic P5 vs. P8 = 0.37, p<0.001; P5 vs. P14 = 0.52, p<0.001; P8 vs. P14 = 0.20, p<0.001; all p-values adjusted for multiple comparisons using Bonferroni correction). Inset shows the process of extracting the angles associated with peak and trough as well as formula for asymmetry index. (C) Power of spindle band oscillations decreases across pup development (n = as in *Figure 3A*; KS statistic P5 vs. P8 = 0.69, p<0.001; statistic P5 vs. P14 = 0.42, p<0.001; P8 vs. P14 = 0.54, p<0.001; all p-values adjusted for multiple comparisons using Bonferroni correction). Inset demonstrates process of extracting spindle band power, from raw trace to filtered trace and extraction of Hilbert power envelope for sample spindle band oscillation from P5 pup. (D) Classification of spindle band events based on their spatial extent and waveform asymmetry for immature (P5–7, green) and mature (P10–14, red) pups. Superimposed line plots demonstrate the marginal probability for each event type (n = 315 events, waveform asymmetry and spatial extent of the marginal probability distributions of immature and mature animals are significantly different; p<0.001). (E) Combinatorial metric of spindle band waveform properties (A: asymmetry; S: spatial extent; P: power) increases nonlinearly across development (black line = mean; shaded blue areas ± SE) with a local minimum at the beginning of the second postnatal week (n = 44 pups). (F) Probability of a local minimum located at each age for datapoints resampled with replacement across days (gray) compared to datapoints resampled with replacement within days (orange), confirming existence of local minimum between P8 and 9.

The online version of this article includes the following figure supplement(s) for figure 4:

**Figure supplement 1.** Model selection and verification of nonlinear developmental trajectory for spindle band oscillation properties in mice.

these results suggest that immature networks lack the capacity to couple discrete frequency bands, whereas after the beginning of the second postnatal week such capacity emerges, potentially signaling a developmental shift in the function of oscillations.

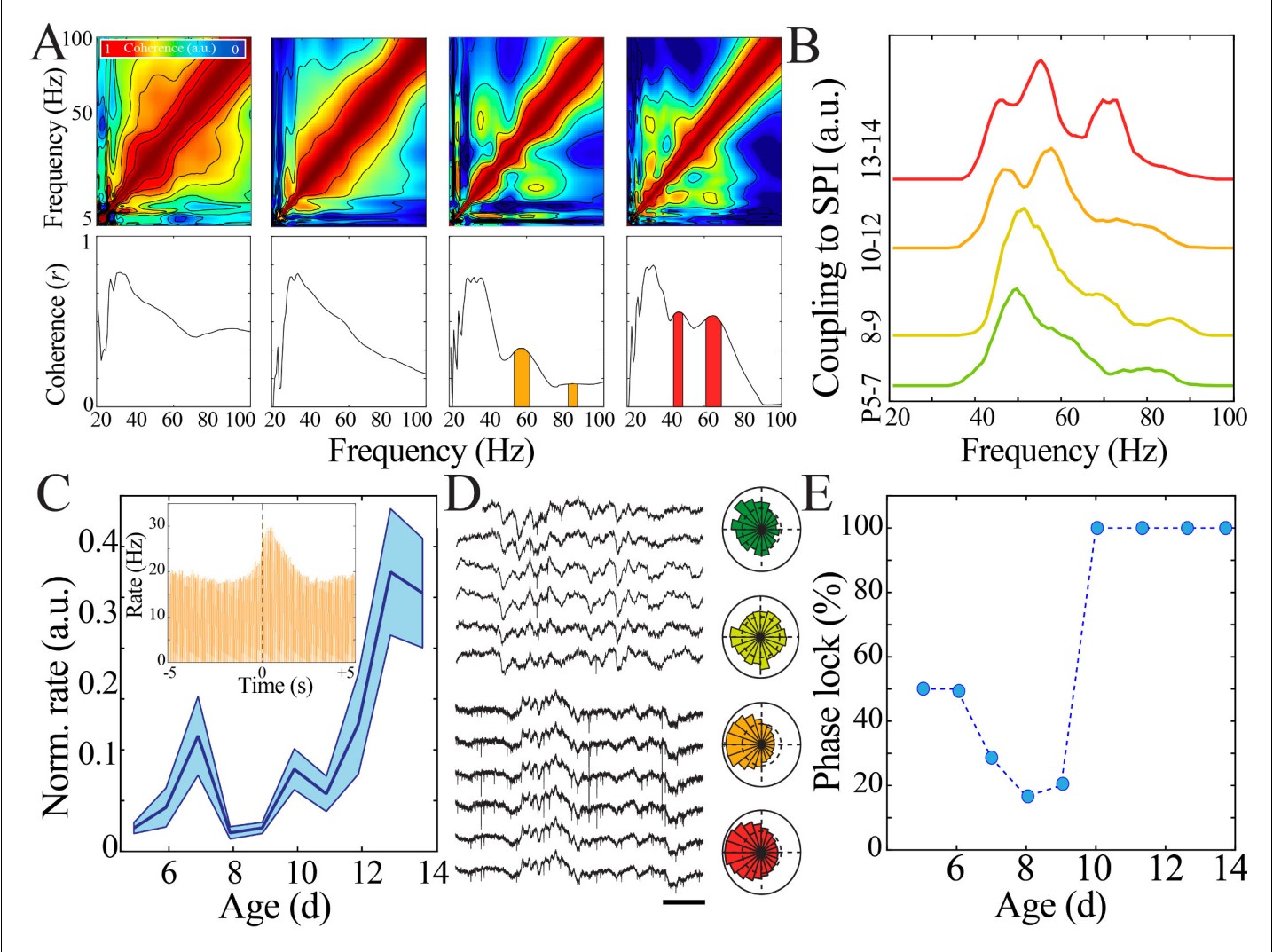

**Figure 5.** Increasing regularity and temporal precision of synaptic activity facilitates recruitment and entrainment of neural spiking across development. (A) Frequency comodulation at the time of spindle band oscillations emerges across development. Comodulograms (upper) and quantification of significant peaks in frequency coherence to spindle band (lower). (B) Significant, discrete peaks of cross-frequency coupling with spindle band oscillations emerge starting at P10 (n = 51 pups). (C) Rate of neural spiking during spindle band oscillation normalized to baseline neural spiking rate increases nonlinearly across development (blue line = mean; shaded blue areas ± SE) with a local minimum at the beginning of the second postnatal week (n = 38 pups). Inset shows sample peristimulus time histogram of neural spiking during spindle band oscillation used for subsequent quantification (starting at t = 0) for a P14 pup. (D) Sample raw time traces of spindle band oscillation and neural spiking from P8 (top) and P13 (bottom) pups. Prominent phase-locking of spikes to trough of spindle band oscillation occurs in P13 pup only. Scale bar 500 ms. Sample polar plots show significant (kappa > 0.1 and Rayleigh p<0.05) phase-locking of neural firing to spindle band oscillations in P5 (green), P11 (orange), and P14 (red) pups, but not at P8 (yellow). (E) Percentage of pups expressing significant (kappa > 0.1 and Rayleigh p<0.05) phase-locking of neural spiking to spindle band oscillations across development (n = 38 pups).

The online version of this article includes the following figure supplement(s) for figure 5:

**Figure supplement 1.** Bootstrapping, model selection and verification of nonlinear developmental trajectory for spiking recruitment to spindle band oscillations in mice.

Furthermore, oscillations facilitate neural communication by biasing spike occurrence to particular phases of the oscillatory waveform. Spindle band oscillations recruited neural spiking above baseline rates to a variable extent across development, with the lowest amount of recruitment occurring at the beginning of the second postnatal week (*Figure 5C*, *Figure 5—figure supplement 1*). This trajectory significantly deviated from linearity with a discrete nadir confirmed by bootstrapping during this epoch (*Figure 5—figure supplement 1A*). P8 and P9 also exhibited the lowest proportion of

recording sessions with significant phase-locking (kappa > 0.1 and alpha < 0.05), and consistency of phase-locking abruptly increased at P10 (*Figure 5D, E*; n = 38 pups implanted with silicon probes). These results reinforce the notion of enhanced information processing capacity in cortical networks after a rapid transitional period at the beginning of the second postnatal week.

Thus, multiple LFP and neural spiking measures followed a similar highly nonlinear developmental trajectory marked by a key transition at the beginning of the second postnatal week in mouse pups. We next aimed to investigate whether this pattern of neural maturation was conserved across species. To accomplish this goal, we analyzed clinically acquired continuous EEG data from 54 human subjects ranging from 36 to 69 weeks post-gestation. EEG recordings were included for analysis only if they were reported as normal by the reading epileptologists and the subject had no known underlying neurological or genetic condition. The majority of subjects received a discharge diagnosis of normal movements/behaviors (e.g., sleep myoclonus) or brief resolved unexplained events (BRUE; *Figure 6—figure supplement 1*). We analyzed epochs of clinically determined quiet/NREM sleep from each subject using data derived from electrodes overlying human somatosensory cortex (parietal or central electrodes; no differences were found between these localizations; *Figure 6—figure supplement 1D*). Raw traces and normalized spectral analysis suggested an epoch of transient neural quiescence between 42 and 47 weeks (*Figure 6A, B*), akin to what we observed at the beginning of the second postnatal week in mice. To quantify these observations, we again used regression fitting to determine whether a nonlinear trajectory was present and bootstrapping of data within and between ages to identify any significant nadirs in neural activity patterns. Wideband power of oscillatory activity followed a highly nonlinear trajectory, with a significant nadir between 42 and 47 weeks (*Figure 6C*, *Figure 6—figure supplement 2A*). Power spectra revealed a lack of prominent frequency peaks during this epoch, as well as an overall steeper decay in power with increasing frequency (corresponding to a larger exponent for the modeled power law; *Figure 6—figure supplement 3A–C*). Quantification of this aperiodic component of the power spectrum across ages similarly revealed a transient increase around the time of nadir in wideband power (*Figure 6—figure supplement 3D, E*). As expected from previous work (*Torres and Anderson, 1985*), duration of oscillatory epochs generally increased over development, but we did observe a period of relative stationarity coinciding with the decrease in wideband power (*Figure 6D*, *Figure 6—figure supplement 2B*). To assess properties of oscillatory activity in these subjects, we detected spindle band oscillations. Such oscillations were present in all ages, consistent with delta brushes in the youngest subjects and sleep spindles in the older subjects (*Figure 6E*). We calculated spatial extent of these oscillations as the co-occurrence of spindle band activity across EEG electrodes and found that this activity shifted from highly localized to broader cortical expression around an inflection point between 42 and 47 weeks (*Figure 6F*, *Figure 6—figure supplement 2C*). We also found evidence for significant cross-frequency coupling of higher frequencies with the spindle band emerging after 47 weeks post-gestation (*Figure 6G, H*; n = 702 spindle band oscillations from 13 subjects; Kruskal–Wallis chi-square for AUC = 138.4; p<0.001). Taken together, these data support the existence of a conserved transient quiescent period in development that bridges a dramatic shift in functional properties of neural networks.

## Discussion

We demonstrate that maturation of cortical network dynamics is preceded by a discrete, relatively quiescent, evolutionarily conserved transition period during early postnatal development. During this period, oscillations and neural spiking display a nadir that signals the shift from local, loosely correlated, putatively sensory-driven patterns to internally organized, spatially distributed, and temporally precise activity. Such a developmental inflection point may reflect a rapid and confluent engagement of mechanisms that enable mature computational processes within cortical networks.

Self-organization in complex systems is often mediated by processes that can facilitate sudden transitions in system behavior. Our results provide evidence for such a transition in developing cortex of rodents and humans. Because rodent neurodevelopment is temporally compressed relative to humans, an inflection epoch that lasts 1–2 days in mouse pups could conceivably extend over weeks in humans (*Clancy et al., 2001*). Prior to this inflection point, oscillatory activity was intermittent with non-uniform cycle-to-cycle waveforms and high broadband frequency coherence. Oscillatory activities after the transition became nearly uninterrupted, exhibiting coupling at discrete frequency

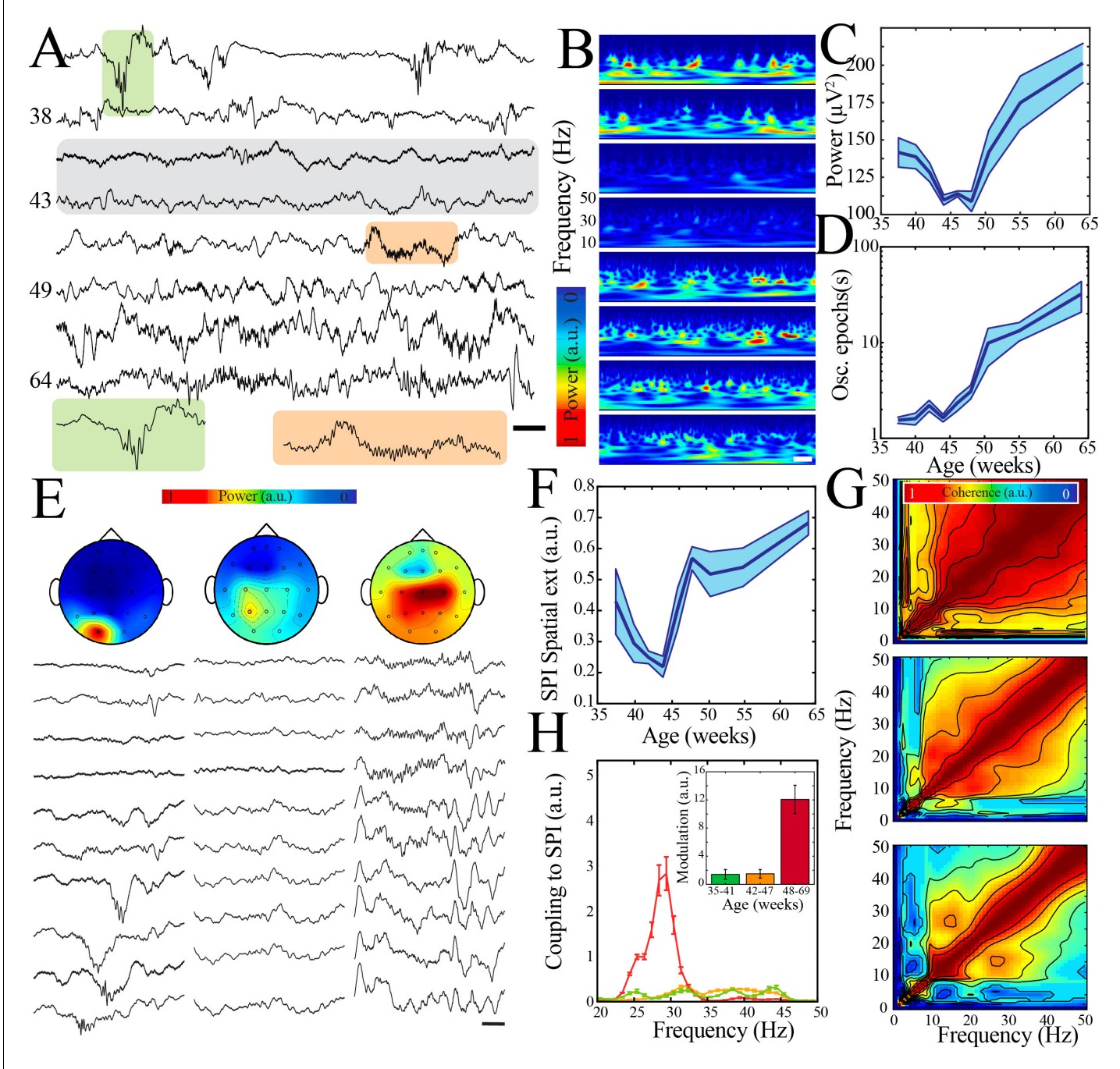

**Figure 6.** A transient quiescent state shifts network dynamics in humans. (**A**) Sample raw traces from human subjects revealing a relative paucity of activity between 42 and 47 weeks during quiet/non-rapid eye movement (NREM) sleep (gray shaded box). Orange shaded box is expanded to reveal characteristic appearance of infantile sleep spindles. Scale bar 1 s. (**B**) Sample spectrograms from human subjects revealing a relative paucity of activity between 42 and 47 weeks during quiet/NREM sleep. Power scale was normalized across sessions. Scale bar 1 s; ages correspond to traces in (**A**). (**C**) Wideband (WB) power changes nonlinearly across development (blue line = mean; shaded blue areas ± SE), with a local minimum between 42 and 47 weeks post-gestational age (n = 54 babies). Inset shows probability of a local minimum located at each age for datapoints resampled with replacement across days (gray) compared to datapoints resampled with replacement within days (orange), confirming existence of local minimum between 42 and 47 weeks. (**D**) Duration of oscillatory epochs increases nonlinearly across development (blue line = mean; shaded blue areas ± SE) with a local minimum between 42 and 47 weeks post-gestational age (n = 54 babies). Inset shows probability of a local minimum located at each age for datapoints resampled with replacement across days (gray) compared to datapoints resampled with replacement within days (orange), confirming existence of local minimum between 42 and 47 weeks. (**E**) Sample raw traces of detected spindle band oscillations across subset of electroencephalography (EEG) electrodes from babies aged 37 weeks (left), 43 weeks (middle), and 58 weeks (right) post-gestational age. Scale bar 1 s. Head models show

*Figure 6 continued on next page*

*Figure 6 continued*

corresponding localization of spindle band power across EEG electrodes in conventional 10–20 placement. Power was normalized across sessions. (F) Spindle band oscillation spatial extent changes nonlinearly across development (blue line = mean; shaded blue areas ± SE), with a local minimum between 42 and 47 weeks post-gestational age (n = 54 babies). Inset shows probability of a local minimum located at each age for datapoints resampled with replacement across days (gray) compared to datapoints resampled with replacement within days (orange), confirming existence of local minimum between 42 and 47 weeks. (G) Frequency comodulation at the time of spindle band oscillations from 36-, 42-, and 56-week post-gestational age subjects. (H) A significant, discrete peak of cross-frequency coupling with spindle band oscillations emerges after 47 weeks (n = 702 spindle band oscillations from 13 subjects). Inset shows quantification of area under the curve for coupling to spindle band oscillations between 25 and 35 Hz.

The online version of this article includes the following figure supplement(s) for figure 6:

**Figure supplement 1.** Demographic characteristics of human subjects.

**Figure supplement 2.** Bootstrapping, model selection and verification of nonlinear developmental trajectory for continuity, wideband power, and spatial extent of spindle band oscillations in human subjects.

**Figure supplement 3.** Spectral features and power law characteristics in human subjects.

bands and increased waveform symmetry. Based on these similarities in oscillatory patterns, we infer that the neural spiking features we observe through invasive recordings in mouse pups could also extend to normally developing human infants, where such recordings are not possible. Neural spiking was more efficiently recruited to oscillatory activity with consistent, strong phase-locking, and higher temporal precision. These features are characteristic of adult neural networks engaged in information processing (*Masquelier et al., 2009*). Thus, a transient period of developmental perturbation may be an evolutionarily conserved indicator of a substantial shift in cortical network operations.

Our results suggest that the cortical network dynamics shift to permit temporally precise neural spiking after the transition period. Significantly synchronized firing within 10–50 ms intervals was absent prior to this period and robustly emerged afterward. This result is consistent with observed increase in pair-wise correlations of spike activity in visual cortex over development (*Colonnese et al., 2017*). Although recruitment of neural spiking during epochs of spindle burst oscillations in immature rodents likely co-activates neurons that share similar afferent inputs (*Khazipov et al., 2004*; *Colonnese et al., 2010*; *Khazipov et al., 2013*; *Winnubst et al., 2015*), more precise coordination is required for establishment of flexible, defined neural assemblies (*Dragoi and Buzsáki, 2006*), and the capacity of circuits to express sequential activation of these assemblies is developmentally regulated (*Farooq and Dragoi, 2019*).

We found that the neural population activity demonstrated correlated suppression over a progressively shorter duration (from approximately 500 to 200 ms) with maturation. This process is in keeping with the progressively decorrelated activity patterns observed when analysis is performed on the time scale of calcium indicators (*Golshani et al., 2009*; *Rochefort et al., 2009*). When oscillations occur in discrete frequency bands, the possibility for complex comodulation arises. This cross-frequency coupling has been linked to information processing in the mature brain, though the precise anatomical substrates and physiological mechanisms that enable its expression are often unknown (*Tort et al., 2009*; *Fujisawa and Buzsáki, 2011*; *Bosman et al., 2012*). We demonstrated that prior to the transient quiescent period the dominant cortical oscillations (centered on spindle band) exhibit high comodulation extending from 1 to 100 Hz. During this period, broadband comodulation abruptly decreased and was subsequently replaced by discrete gamma oscillations. Therefore, the oscillatory infrastructure to support cross-frequency coupling arises concomitantly with fine time-scale neural spiking synchronization and robust phase-locking of these spikes to LFP, suggesting an emergence of adult-like information processing capacity at this developmental timepoint.

Despite sharing a peak frequency, immature and mature spindle band oscillations are highly differentiable on the joint basis of waveform, power, and spatial extent. Correspondingly, the cellular mechanisms that permit expression of these oscillations, as well as their functional purpose, are different. Comparison of oscillatory features can shed light on the functional capacity of the network. We found that although of high amplitude, spindle band frequency oscillations in the most immature mouse pups were highly and uniformly confined to localized regions of cortex (200–500 μm) roughly corresponding to cortical columns (*Yang et al., 2013*; *Feldmeyer et al., 2013*). Similarly, human delta brushes exhibited restricted localization even given the limited spatial resolution of clinical EEG. During the transition period, spindle band oscillations in mouse pups had higher spatial extent,

but markedly reduced amplitude, perhaps facilitated by increased intracortical synaptogenesis and maturation of corticothalamic afferents (*Naskar et al., 2019*). The transition period in humans was marked by transiently decreased spatial extent of spindle band oscillations, which we hypothesize is related to attenuation of low-power activity by the intervening tissue between cortical surface and scalp electrode. We observed that mature spindle band activity across species was characterized by increased and customizable spatial extent. Thus, the end of the transient quiescent period marks capacity for oscillatory frequencies that were previously confined to local processing units to extend into broader cortical areas with the potential to establish intercortical communication.

Our study was limited to somatosensory cortex, and the existence of a transient quiescent epoch outside of this brain region is unknown. Similar to our results in the spindle band, early gamma oscillations disappear around the beginning to the second postnatal week, to be later replaced with gamma rhythms characteristic of adult animals (*Minlebaev et al., 2011*). In visual cortex, a transient nadir in continuity was not identified, but continuity did rise sharply after P8 with a nonlinear trajectory (*Shen and Colonnese, 2016*). Furthermore, the least synchronous neural spiking activity in visual cortex occurred at the beginning of the second postnatal week, potentially in keeping with our decrease in temporal precision of spiking at this time (*Colonnese et al., 2017*). Higher order cortices, which do not directly receive sensory thalamic inputs, may express a different developmental trajectory, in keeping with their distinctive role in information processing, though oscillatory expression during this same epoch has been identified as critical for intact adult function (*Bitzenhofer et al., 2015*; *Bitzenhofer et al., 2021*; *Bitzenhofer et al., 2020*). Therefore, multiple lines of evidence support an abrupt change in cortical network function at the beginning of the second postnatal week. It is possible that our use of surface arrays, which sample summated LFP activity from the undisturbed superficial cortical layers (I–III), highlights the transient quiescent state compared to penetrating probes that disrupt the cortical surface upon implantation. Furthermore, our conducting polymer-based electrodes have lower impedance than the silicon probes used in most neonatal rodent studies (*Khodagholy et al., 2015*), potentially increasing the sensitivity to changes in oscillatory power and continuity.

Given the multitude of anatomic and functional changes that occur during the early postnatal epoch, it is likely that the shift in network operating mode we observe is mechanistically multifactorial. A potential driver for this state is maturation of functional cortical inhibition. Functional inhibitory connections regulate physiological oscillations in mature neural networks enabling submillisecond precision of spike timing and facilitating inter-regional synchronization (*Buzsáki and Chrobak, 1995*; *Cobb et al., 1995*; *Pouille and Scanziani, 2001*). In contrast, interneurons do not substantially pace immature oscillations during the first postnatal week, though they can be recruited to and contribute to spatial properties of this activity (*Khazipov et al., 2004*; *Minlebaev et al., 2011*). Functional feedforward inhibition rapidly matures around the beginning of the second postnatal week in mice, resulting in emergence of adult-like hyperpolarizing synaptic potentials and precisely timed action potentials in glutamatergic neurons within the microcircuit (*Daw et al., 2007*). This timepoint, derived from in vitro data, temporally overlaps with the onset of adult-like neural network activity we observe in vivo. In the human brain, when mature functional inhibition arises is not known, but ongoing migration of cells that differentiate into interneurons in the months after birth (*Paredes et al., 2016*) suggests the possibility of a postnatal transition to such network activity. The aperiodic component of the power spectrum has been linked to excitatory-inhibitory balance (*Gao et al., 2017*), and our finding that this value transiently increases during the quiescent period could support a temporary increase in inhibition in the human brain. Such a mechanism could underlie the known postnatal disappearance of the EEG bursts characteristics of trace alternant (*Ellingson and Peters, 1980*), resulting in the observed low-voltage state with lack of organized oscillatory activity prior to emergence of mature NREM sleep patterns.

Whether a period of quiescence is required for physiological transition to mature cortical processing remains unclear. Manipulations that could modulate its expression, such as alteration of the neuromodulatory milieu, activation/suppression of specific cell-type activity, or modulation of the sensory environment (*Smith et al., 2018*), could address this question. Forging linkages to activity patterns observed using calcium imaging (*Wilson et al., 2018*) and enabling high spatiotemporal resolution chronic in vivo electrophysiology will also be critical. In humans, neuronal dynamics suggestive of mature information processing mechanisms are established by 6 months of age (*Jannesari et al., 2020*), and changes in sleep patterns have been identified as markers for brain

connectivity from toddlers to adolescents (*Kurth et al., 2017*). Extending such measures to earlier timepoints could help to clarify functional ramifications of the quiescent period in humans. Furthermore, the manifestation of a developmental epoch of relative quiescence in electrophysiological activity suggests the possibility for use as a network-level biomarker that could be altered in disorders of neurodevelopment, from mouse models to human subjects.

Neural processes in the adult brain are characterized by a specific constellation of network dynamics that are necessary for execution of complex behaviors and cognition (*Tolner et al., 2012*). Our results suggest that the capacity for these processes simultaneously emerges abruptly after a distinct transition period characterized by relative neural quiescence. This transient quiescent period may be a critical stage that indicates a shift in the neural network operating mode from a predominantly external stimulus-driven state to one that facilitates the internal representations associated with planning and executive function.

# Materials and methods

## Key resources table

| Reagent type (species) or resource | Designation | Source or reference | Identifiers | Additional information |
|---|---|---|---|---|
| Strain, strain background (*Mus musculus*) | SW (Crl:CFW) | Charles River | IMSR Cat# CRL:024, RRID:IMSR_CRL:024 | Mouse |
| Antibody | Anti-VGLUT2 (polyclonal; target: mouse, rat; host: guinea pig) | Synaptic Systems | Cat# 135 404, RRID:AB_887884 | Primary antibody, dilution 1:1000 |
| Antibody | 594 Donkey Anti-Guinea Pig IgG (H+L) (polyclonal; target: guinea pig; host: donkey) | Jackson ImmunoResearch Labs | Cat# 706-585-148, RRID:AB_2340474 | Secondary antibody, dilution 1:500 |
| Software, algorithm | MATLAB | MathWorks | RRID:SCR_001622 | |
| Software, algorithm | Chronux | Chronux.org | RRID:SCR_005547 | |

## Probe fabrication

PEDOT:PSS (Clevios PH1000) was purchased from Heraeus. Ethylene glycol, (3-glycidyloxypropyl)trimethoxysilane (GOPS), 4-dodecyl benzene sulfonic acid (DBSA), and 3-(trimethoxysilyl)propyl methacrylate (A-174 silane) were purchased from Sigma-Aldrich. Micro-90 concentrated cleaning solution was purchased from Special Coating Services. AZnLOF2020 (negative photoresist), AZ9260 (positive photoresist), and AZ 400K and AZ 300MIF (metal ion free) developers were acquired from Micro-Chemicals, Merck. To create the PEDOT:PSS films, a mixture of aqueous dispersion (Clevios PH1000) and ethylene glycol was prepared and mixed with GOPS (1 wt%) and DBSA (0.1 wt%). The fabrication process involved deposition and patterning of parylene C, evaporation of Au for electrodes and interconnects, and PEDOT:PSS films coating. Parylene C (1.2-µm-thick) was deposited on quartz wafers (100 mm outer diameter [O.D.], thickness of 1 mm) using an SCS Labcoater 2. For the metal lift-off, AZnLOF2020-negative photoresist was spin-coated at 3000 r.p.m. on the substrate, baked at 105°C for 90 s, exposed to ultraviolet light using a Suss MA6 Mask Aligner, and developed with AZ 300MIF developer. Metallic layers (10 nm Ti, 150 nm Au) were deposited using an e-beam metal evaporator (Angstrom EvoVac Multi-Process) and patterned by soaking the substrate in a bath of resist remover. A second layer of parylene C (insulation layer) was deposited to a thickness of 1.2 µm and its adhesion to the bottom layer was enhanced by the addition of 3-(trimethoxysilyl)propyl methacrylate (A-174 silane) during chemical vapor deposition. An anti-adhesion agent (5 wt% Micro-90 diluted in deionized water) was deposited on the substrate followed by an additional sacrificial layer of parylene C that was subsequently used for the peel-off process. The stacked layers were patterned with a layer of AZ9260-positive photoresist and dry etched with a plasma-reactive ion-etching process (Oxford Plasmalab 80; 180 W, 50 sccm $O_2$, and 2 sccm $SF_6$) to shape electrode area and electrical contact pads. Specifically, AZ9260 was spin-coated at 5000 r.p.m., baked at 115°C for 90 s, exposed using a Suss MA6 Mask Aligner, and developed with AZ400K developer (1:4 with deionized

water). To obtain clean etching of large areas, the electrical contact pads were guarded by an extra layer of AZnLOF2020 (3000 r.p.m.) between the metal layer and the silane-treated parylene. Each pad was etched along its perimeter, and the parylene deposited on top of the photoresist was removed once immersed in acetone bath. Finally, PEDOT:PSS was spin-coated on top of the electrodes and patterned by peeling off the last parylene layer. The probes had 119 electrodes regularly spaced on a diagonal square centered lattice with pitch of 77–177 µm and regularly interspersed perforations were created. Total device thickness was 4 µm.

## Animal surgical procedure

All animal experiments were approved by the Institutional Animal Care and Use Committee at Columbia University Irving Medical Center. Data from 108 Swiss-Webster mouse pups (3–9 g, 5–14 days of age) that underwent intracranial implantation was used for neurophysiological analysis. Pups were kept on a regular 12 hr–12 hr light-dark cycle and housed with the mother before surgery. No prior experimentation had been performed on these mice. Anesthesia was induced by hypothermia in ice for P5–6 pups and with 3% isoflurane for pups aged P7–14. Pups were then comfortably placed on a customized platform for head fixation, surgery, and recording. Acclimatization to head fixation was not performed as it has been found to be unnecessary for neonatal rodents; careful positioning and 'swaddling' was performed to maximize pup comfort (*Blumberg et al., 2015*; *Corner and Kwee, 1976*). All animals were maintained under anesthesia with 0.75–1.5% isoflurane during surgery. Time under anesthesia was minimized to facilitate postoperative recovery. Pre- and postoperative systemic and local analgesics were employed, including subcutaneous bupivacaine, systemic ketoprofen, and topical lidocaine. These analgesics were selected to attain appropriate pain control while minimizing effects on neurophysiological activity in the postoperative period (*Blumberg et al., 2015*; *Foley et al., 2019*). Electrodes fabricated from fine gauge wire were inserted into the subcutaneous tissue of the neck and abdomen. These electrodes continuously registered electrocardiography (ECG) and electromyography (EMG) signals throughout the surgical and recording period, and were used to monitor recovery from anesthesia. A custom metal plate was attached to the skull to provide head fixation. For NeuroGrid implantations, a craniotomy was centered over primary somatosensory cortex and the array was placed over the dura. For silicon probe implantations, H32 NeuroNexus 32-site single-shank probes were used. A small cranial window over primary somatosensory cortex (approximately AP −1.5 mm, ML 2 mm; adjusted proportionally based on pup age) was opened, and the probe was stereotactically advanced until all electrodes were inserted below the pial surface (900 µm). Postoperatively, the pup was comfortably positioned, provided with familiar olfactory cues, and placed in a custom recording box with temperature and humidity control. Temperature was regulated by a heater equipped with feedback control to ensure stable, physiological body temperature of the pups throughout the recording period.

## Rodent neurophysiological signal acquisition

Neurophysiological signal acquisition was performed in the custom recording box, which was grounded to shield ambient electrical noise. Concurrent video monitoring was performed using a camera mounted inside the box. Pups recovered from anesthesia, as determined by recovery criteria including increase of heart rate to plateau and onset of sleep-wake cycling, for a minimum of 30 min before neurophysiological recordings were used for analysis. The recovery period was prolonged until recovery criteria were met, and all pups used for experimentation recovered within 60 min postoperatively. Neurophysiological signals were amplified, digitized continuously at 20 kHz using a local preamplifier (RHD2000 Intan Technology), and stored for offline analysis with 16-bit format. Spontaneous activity was acquired for 1–2 hr prior to euthanasia and perfusion.

## Perfusion and histology

At the end of experimentation, pups were euthanized with an overdose of pentobarbital (100 mg/kg). For the NeuroGrid experiments, the position of the array was visualized using light microscopy and the brain tissue underlying the corners of the array was stereotactically marked using a 30 gauge needle coated with chitosan compound (*Rauhala et al., 2020*). Pups were perfused with phosphate-buffered saline (PBS), followed by 4% paraformaldehyde. Brains were stored for 24–48 hr in 4% PFA at 4°C prior to transfer to PBS solution. Cortices were flattened at room temperature (RT) after

removal of the underlying midline structures and kept in 4% PFA for 24 hr (4°C). Flattened cortices were sliced using a Leica vibratome vt1000S at 50 μm. The slices were then incubated for 1 hr at RT in a blocking solution composed of PBS (0.01 M), 3% Normal Donkey Serum, and 0.3% Triton X-100. Slices were transferred to a solution containing anti-Vesicular Glutamate Transporter 2 (VGlut2) polyclonal guinea-pig antibody (AB2251-I Sigma Millipore) in a 1:2000 dilution and kept overnight at 4°C. Subsequently, the tissue was washed and incubated with secondary antibody, Alexa Fluor 594 AffiniPure Donkey Anti-guinea pig IgG (H+L) (706-585-148) from Jackson Immunoresearch at 1:1000 dilution. Lastly, the tissue was incubated with DAPI (D9542-5MG; Sigma-Aldrich) at 0.2 μg/mL and after washing, the brain slices were mounted with Fluoromount-G (00-4958-02, ThermoFisher). Fluorescent images were captured and tiled using a computer-assisted camera connected to an ECHO Revolve microscope. The images were adjusted for brightness and contrast and assembled into panels using ImageJ and Adobe Illustrator. The regularly spaced, hexagonal lattice of electrode spacing across the NeuroGrid enabled localization of each electrode relative to the marked corners of the array and primary sensory cortices. The spatially extensive nature of the NeuroGrid relative to pup brain size facilitated consistent anatomical targeting. Localization of silicon probe electrodes to cortical region and layers was performed using a combination of intraoperative visualization and histological reconstruction. Probes were inserted under visual guidance, with the most superficial electrode placed just below the pia. Regular, precise spacing of electrodes along the shank enabled determination of each electrode's depth from the cortical surface. Coronal brain slices stained with DAPI and vGlut2 permitted localization of probe implantation site relative to barrel cortex.

## Rodent neurophysiological signal preprocessing and state scoring

All recordings were visually inspected for quality, and channels that exhibited ongoing electrical or mechanical artifact were removed from analysis. Data was analyzed using MATLAB (MathWorks) and visualized using Neuroscope. Raw recordings were downsampled to 1250 Hz to obtain LFPs, and band-pass filtered between 250 Hz and 2500 Hz for analysis of neural spiking. Signals from EMG wires inserted into nuchal and ventral subcutaneous tissue were high-pass filtered at 300 Hz, rectified, and smoothed to obtain a power envelope. The mean of this high-pass filtered power was determined. Epochs consistent with high tone were identified when the signal surpassed a threshold defined by 1.5–3.5 standard deviations above this mean. Wakefulness was defined as a period of sustained high tone for at least 1.5 s. Myoclonic twitches involving contractions of the muscles of the head, forelimbs, hindlimbs, or trunk as observed on video recording were correlated with an increase in EMG signal that surpassed the threshold for < 1.5 s. Muscle atonia (consistent with sleep) was identified when the EMG signal remained below the threshold for at least 2 s. Sleep epochs were further classified as putative active or quiet sleep. Quiet sleep was identified when muscle atonia was paired with quiescence of behavior for at least 10 consecutive seconds. Active sleep was identified by myoclonic twitches that occurred on a background of muscle atonia. Similar approaches have been shown to correlate with electrophysiological markers of active and quiet sleep, and have been used for state-dependent developmental analyses (*Seelke and Blumberg, 2008*; *Turek, 1999*).

## Human neurophysiological signal prepreprocessing and state scoring

We retrospectively analyzed EEG recordings from 54 patients who underwent continuous monitoring with surface EEG as part of clinical diagnostic assessment. Analysis of these data was approved by the Institutional Review Board at Columbia University Irving Medical Center, and all data collection occurred at this institution. All data reviewed was initially obtained for clinical management purposes and informed consent was waived as per 45 CFR 46.116. Each patient had EEG electrodes placed based on the internationally recognized 10–20 configuration. Electrode impedance was monitored and maintained within appropriate ranges as per American Clinical Neurophysiology Society Guidelines by certified clinical EEG technologists. Patients ranging in age from birth to 8 months who had at least 4 hr of high-quality continuous EEG monitoring were identified through the clinical electrophysiology database (Natus). Using the hospital admission record, patients were excluded if (1) the EEG was reported as abnormal by the reading electrophysiologist; (2) the patient had a known underlying neurological or genetic condition; (3) the patient was diagnosed with a new neurological condition during the course of the hospital admission; and (4) corrected gestational age could not

be determined from information contained in the medical record. Indication for EEG in eligible patients was most commonly paroxysmal movements concerning to parents for abnormal activity, with discharge diagnosis identifying normal movements/behaviors (e.g., sleep myoclonus, roving eye movements during REM sleep) or BRUE (*Figure 6—figure supplement 1*). Corrected gestational age, rounded to the nearest week, was used to classify patient age. If the phrase 'full term' was employed in the medical record, corrected gestational age was calculated based on birth at 40 weeks. This approach allowed a precision of 2 weeks in corrected gestational age. Recordings always included both waking and sleep epochs. Three to five representative epochs of artifact-free quiet/NREM sleep were identified for subsequent analysis by a certified clinical electrophysiologist. Data was sampled at 256–512 Hz by clinical amplifiers. Raw data (referential) exported from Natus was used for all analyses. If electrical noise contaminated the recording, a 60 Hz notch filter was used. P3 or P4 electrodes, overlying the parietal lobe, were selected for spectral analysis. If these electrodes were non-functional, C3 or C4 electrodes, overlying the central sulcus, were used. No differences were found when using data derived from parietal and central electrodes in subjects where all channels were functional.

## Spectral analysis and oscillatory activity

To visualize spectrograms, a parametric autoregressive (order of 1) whitening method was first applied to the data. For rodent data (sampled at 1250 Hz) and human data (sampled at 256–512 Hz), time-frequency decomposition was performed using analytic Gabor wavelet transform to enhance temporal resolution. Wavelet spectrograms centered at twitch times were normalized by z-scoring and averaged to obtain trigger-averaged spectrograms. Power was extracted for quantification from the magnitude of the analytic Gabor wavelet transform. Epochs of oscillatory activity were defined by presence of rectified signal amplitude above the 99th percentile of the wideband noise floor. To measure periodic and aperiodic properties of human neural data, power spectra were extracted using multi-taper method (http:/chronux.org) and modeled using 'fitting oscillations and one over f' (*Corner and Kwee, 1976*).

## Modeling neural properties across developmental trajectories

In order to quantify the changing tendency of each feature over time, we modeled the trends with linear and polynomial regression models. To find the optimal fit that most accurately represented the overall trend without overfitting, we used LOOCV to evaluate the models. The model with the minimum MSE was considered to have the best fit. When model fit predicted existence of local extrema, we used bootstrapping to estimate the location of these extrema for the trajectory of each property we investigated. For animal data, we resampled with replacement within each day from P5 to P14. For human data, we resampled within each 1–2 weeks. Age was jittered with a window of 0.5 days and 0.5 weeks for animal and human data, respectively, to aid with fitting. We found the age corresponding to the first peak or trough of the regression line across development. This process was repeated 10,000 times to generate a probability distribution of location of local extrema across the developmental trajectory. The same process was repeated using resampling with replacement across all timepoints of the trajectory (10,000 iterations) to generate a null distribution for comparison.

## Spindle band oscillation detection

Spindle band oscillations were detected based on wavelet-derived power and duration parameters. Intervals of oscillatory activity containing spindle band power were first detected and a ratio of normalized autoregressive wavelet (Gabor)-based $P_{AR}$ was then calculated to identify discrete spindle band oscillations. Oscillatory intervals were detected by comparing the spindle band power ($P_{spi}$) of 8–25 Hz with threshold. Thresholds were determined relative to median absolute deviation ($\frac{median(|Pspi|)}{0.6745}$) in the sleep intervals. Because this value varied markedly across development, thresholds were derived empirically but kept consistent across each age group. The value was decreased for silicon probe recordings in P5–7 pups compared to NeuroGrid recordings to maintain consistent detection despite quantifiably different signal to noise ratio. $P_{AR}$ was then calculated based on the following equation:

$$P_{AR} = \frac{P_{spi} - \left(P_{low} + P_{high}\right)}{P_{spi} + \left(P_{low} + P_{high}\right)}$$

where spindle band power ($P_{spi}$) was based on 8–25 Hz, low band power was based on 1–5 Hz ($P_{low}$), and high band power was based on 30–80 Hz ($P_{high}$). Spindle band events were identified when the ratio crossed –0.1 for a minimum of 300 ms and a maximum of 5 s and had a peak greater than 0. All detections were visually inspected for accuracy for each recording session and independently verified by an analyst blinded to detection parameters.

In human EEG data, spindle band oscillations were detected across all channels as in mice, using a constant threshold relative to median absolute deviation ($\frac{median(|Pspi|)}{0.6745}$) in the intervals of quiet/NREM sleep and $P_{AR}$ was calculated as above. For subjects aged less than 40 weeks, the low band power was based on 2–6 Hz to aid with accurate detection of delta brushes. The channel with maximal oscillation power in the spindle band was selected for further analysis.

## Spindle characterization

Spindle band oscillations were characterized by their power, spatial extent, and spindle asymmetry (*Cole and Voytek, 2017*). Power was calculated by bandpass filtering each spindle band oscillation between 8 and 25 Hz. We then calculated the rectified value of Hilbert transformed filtered signals to derive the power envelopes. The median value of each spindle band oscillation was defined as its power. Spatial extent was calculated by determining the number of channels simultaneous expressing spindle band oscillations with a manually selected channel located in primary somatosensory cortex for mice and the channel with the highest spindle band power for humans (intersection of spindle start and end points > 300 ms and separation of spindle midpoints $\leq$ 50 ms on channels being compared). The spatial extent was expressed as a ratio of the number of channels with simultaneous occurrence and the total number of functional channels on the NeuroGrid. Data used for these calculations had a similar proportion and placement of channels located within primary somatosensory cortex. To quantify the sharpness of each spindle band oscillation's peak and trough, they were first broadly bandpass filtered at 5–30 Hz. Rising and falling zero-crossing points were identified. Peaks were recognized as the maximum timepoint in raw data between a rising zero-crossing point and a falling zero-crossing point; troughs were recognized as the minimum timepoint in raw data between a rising zero-crossing point and a falling zero-crossing point. Sharpness of a peak was defined as the mean difference between the voltage at the oscillatory peak $V_{peak}$ and the voltage at timepoints before ($V_{before}$) and after the peak ($V_{after}$), where the timepoint corresponded to a phase shift of approximately π (8 ms). Trough sharpness was defined similarly:

$$Sharp_{peak} = \frac{\left(V_{peak} - V_{before}\right) + \left(V_{peak} - V_{after}\right)}{2}$$

$$Sharp_{trough} = \frac{\left(V_{before} - V_{trough}\right) + \left(V_{after} - V_{trough}\right)}{2}$$

As the absolute difference between the extrema and the surrounding timepoints increases, the extrema sharpness increases. Extrema mean sharpness ratios (ESR) were calculated as a metric to quantify the sharpness asymmetry of each spindle. The ESR was defined in the following manner:

$$ESR = \frac{\frac{1}{N_{peaks}}\sum_{peaks} sharp_{peak}}{\frac{1}{N_{troughs}}\sum_{troughs} sharp_{trough}}$$

Larger deviation from 1 indicated greater sharpness asymmetry of the spindle.

## Comodulogram and cross-frequency coupling

Spectral analysis of the data was performed, and any data contaminated with electrical noise (60 Hz) was eliminated from analysis to prevent spurious correlations. Comodulograms of cross-frequency coupling were computed based on a Gabor wavelet of raw data around the time of spindle band oscillation (4 s), followed by a Gaussian window convolution with duration matched to the frequency.

The correlation matrix was calculated based on the magnitude of the wavelet components. Comparisons with p<0.05 after Bonferroni–Holm correction for multiple comparisons were considered statistically significant. Columns of the correlation matrix corresponding to spindle band (10–20 Hz) were extracted and summated to obtain a curve representing the magnitude of frequency coupling to this band. Each local maximum coupling value was detected along with its peak prominence (how significant a peak is on account of its intrinsic value and location relative to other peaks) and peak width (frequency range with between half-prominence points). Peaks were thresholded based on prominence and width ($\Delta f < 1/f_c$, peak prominence > 0.75). Wide-band coupling index was calculated by (peak value $\times$ peak prominence)/peak width. AUC was calculated by integrated the summated coupling curves.

## Neural spiking analysis

Neural spiking was detected on each channel by thresholding negative peaks greater than four times the high-pass filtered noise floor during epochs of sleep. Coincident spike times across a greater distance than expected for physiological spike waveforms (~300 μm) were presumed to be non-physiological and eliminated from all channels. Electrodes were allocated based on histology and electrophysiological characteristics into zones roughly corresponding to superficial (I–III) and deep (IV–VI) cortical layers. Electrodes from two non-contiguous sites (>200 μm vertical separation) within the deep cortical layers and one site from superficial cortical layers were used for subsequent analysis. Because individual neuron action potentials could potentially be detected on more than one channel, spikes occurring on different channels within the same histological zone < 2 ms apart were presumed to be duplicate detections and only the first spike was used for analysis. ISIs for the neural population were calculated as the absolute time between sorted spike times within the zone. A modified convolution method was used to determine 95% confidence intervals for population spike time autocorrelograms with a bin size of 10 ms. Peristimulus time histograms were calculated using a bin size of 50 ms with 95% confidence intervals determined from a shuffled distribution of spike times. To derive spike phase-locking to spindle band oscillations, the oscillatory epochs were first downsampled to 125 Hz and narrowly bandpass filtered at 9–16 Hz. Phase was extracted using Hilbert transform. Phase bins π/24 were used for quantification of spike phase preference.

## Statistics

Statistical analysis was performed using a combination of open-source MATLAB toolboxes and custom MATLAB code. A modified convolution method was used to determine 95% confidence intervals for correlograms. Significance of phase-locking to spindle band oscillations was determined using alpha < 0.05 with a corresponding kappa value of > 0.1 Kuiper test (Circular Statistics Toolbox for MATLAB). Null distributions were generated based on 500 instances of data shuffling and used to calculate 95% confidence intervals. Probability distributions were compared using two-sample Kolmogorov–Smirnov tests with correction for multiple comparisons. Differences between groups were calculated using non-parametric rank-sum (Wilcoxon) or ANOVA (Kruskal–Wallis with Bonferroni correction) depending on the nature of the data analyzed. When event size varied between groups, a random sample of events was selected from groups with larger event sizes to ensure group differences were not driven by degrees of freedom. Error bars represent standard error of mean. Testing was two-tailed and significance level was p<0.05.

## Acknowledgements

This work was supported by the Department of Neurology and Institute for Genomic Medicine at Columbia University Irving Medical Center as well as the School of Engineering and Applied Science at Columbia University. The device fabrication was performed at (1) Columbia Nano-Initiative, (2) Cornell NanoScale Facility (CNF), a member of the National Nanotechnology Coordinated Infrastructure (NNCI), which is supported by the National Science Foundation (grant ECCS-1542081). This project has received funding from the European Union's Horizon 2020 research and innovation program under the Marie Skłodowska-Curie grant agreement no. 799501. This work was supported by the NSF CAREER (1944415), NIH R21 EY032381, Columbia School of Engineering, as well as the Department of Neurology and Institute for Genomic Medicine at Columbia University Irving Medical Center. Human subject data was acquired through New York Presbyterian Hospital. Thanks to the

Churchland group and Daniel Levenstein for fruitful methodological discussion. We thank all Gelinas, Khodagholy, Buzsaki, and Fishell laboratory members for their support.

## Additional information

### Funding

| Funder | Grant reference number | Author |
| --- | --- | --- |
| National Institutes of Health | R21 EY032381 | Dion Khodagholy Jennifer N Gelinas |
| H2020 European Research Council | 799501 | Soledad Dominguez |
| NSF | 1944415 | Dion Khodagholy |

The funders had no role in study design, data collection and interpretation, or the decision to submit the work for publication.

### Author contributions

Soledad Domínguez, Conceptualization, Data curation, Formal analysis, Validation, Investigation, Visualization, Methodology, Writing - original draft; Liang Ma, Data curation, Formal analysis, Validation, Investigation, Visualization, Methodology, Writing - original draft, Writing - review and editing; Han Yu, Software, Formal analysis, Validation, Investigation, Visualization, Methodology, Writing - original draft, Writing - review and editing; Gabrielle Pouchelon, Conceptualization, Validation, Investigation, Visualization, Methodology; Christian Mayer, Conceptualization, Data curation, Validation, Visualization, Methodology; George D Spyropoulos, Claudia Cea, Data curation, Visualization, Methodology; György Buzsáki, Gordon Fishell, Conceptualization, Writing - review and editing; Dion Khodagholy, Conceptualization, Resources, Data curation, Software, Formal analysis, Supervision, Funding acquisition, Validation, Investigation, Visualization, Methodology, Writing - original draft, Project administration, Writing - review and editing; Jennifer N Gelinas, Conceptualization, Resources, Data curation, Formal analysis, Supervision, Funding acquisition, Validation, Investigation, Visualization, Methodology, Writing - original draft, Project administration, Writing - review and editing

### Author ORCIDs

Soledad Domínguez  https://orcid.org/0000-0003-0378-2442
Liang Ma  https://orcid.org/0000-0003-3883-1902
Han Yu  https://orcid.org/0000-0002-7110-7716
György Buzsáki  http://orcid.org/0000-0002-3100-4800
Gordon Fishell  http://orcid.org/0000-0002-9640-9278
Jennifer N Gelinas  https://orcid.org/0000-0002-1164-638X

### Ethics

Human subjects: We retrospectively analyzed EEG recordings from 54 patients who underwent continuous monitoring with surface electroencephalography (EEG) as part of clinical diagnostic assessment. Analysis of these data were approved by the Institutional Review Board at Columbia University Irving Medical Center, and all data collection occurred at this institution. All data reviewed was initially obtained for clinical management purposes and informed consent was waived as per 45 CFR 46.116.

Animal experimentation: All animal experiments were performed in strict accordance with the recommendations in the Guide for the Care and Use of Laboratory Animals of the National Institutes of Health and approved by the Institutional Animal Care and Use Committee at Columbia University Irving Medical Center, protocol AABI5568.

### Decision letter and Author response

Decision letter https://doi.org/10.7554/eLife.69011.sa1
Author response https://doi.org/10.7554/eLife.69011.sa2

## Additional files

### Supplementary files
• Transparent reporting form

### Data availability
Source data are presented in Figure supplements and uploaded to Dryad. Data pertaining to human subjects is governed by IRB policy and can be accessed through application to the IRB. Pooled, processed human subject data are uploaded to Dryad.

The following dataset was generated:

| Author(s) | Year | Dataset title | Dataset URL | Database and Identifier |
|---|---|---|---|---|
| Khodagholy D | 2021 | A transient postnatal quiescent period precedes emergence of mature cortical dynamics | https://doi.org/10.5061/dryad.15dv41nxp | Dryad Digital Repository, 10.5061/dryad.15dv41nxp |

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
