## [Decision Letter]

Thank you for submitting your article "A transient postnatal quiescent period precedes emergence of mature cortical dynamics" for consideration by *eLife*. Your article has been reviewed by 3 peer reviewers, including Sacha B Nelson as the Reviewing Editor and Reviewer #1, and the evaluation has been overseen by Tirin Moore as the Senior Editor. The following individual involved in review of your submission has agreed to reveal their identity: Matthew T Colonnese (Reviewer #3).

Essential revisions:

The following list of essential revisions were developed after rather extensive discussion between the reviewers.

1) Demonstrate more thoroughly that the pausing phenomenon is not an artifact by:

a) quantify twitching and sleep state distributions across ages to show that the 'quiescence' is not a simple function of less movement or active sleep.

b) quantify the spiking from superficial layers (2-4 or 2-3) since spiking data from layers, 4-6 are used, but these layers are both very diverse in their behavior, and the least likely to be strongly correlated with spindle-bursts (maximal in layer 2-4)..

c) Report all the surgical details as requested by Reviewer #2 in the Recommendations for the Authors. Present an analysis of recovery time for neural activity (ECOG or spike rate) showing that differences in surgical recovery between ages cannot account for the quiescent period.

2) Textual changes to present a more balanced view of the unknown function of the "pause" and to temper or eliminate claims of causality that are unwarranted.

*Reviewer #1 (Recommendations for the authors):*

There are places where the authors presume causality without evidence. For example, although twitches are clearly accompanied by changes in LFP power, it is not clear whether the changes in LFP power cause the twitches or vice versa or whether both are caused by some additional unobserved phenomenon. Similarly, Figure 4 demonstrates developmental changes in cross-frequency coupling. While it is reasonable to hypothesize that this is due to changes in the regularity and temporal precision of synaptic activity, this has not been tested, and other changes, such as changes in the intrinsic properties of neurons could also contribute, or could even make a greater contribution.

The discussion speculates that the pause, or distinct transition period itself plays a functionally important role, but little evidence is put forward that this is more likely than that the pause is an epiphenomenon of shifting properties of thalamocortical circuits. The balance of the discussion could be improved by acknowledging this possibility. This might enhance the impact by interesting others in experiments to test the functional role by perturbing it.

*Reviewer #2 (Recommendations for the authors):*

1. It is not clear to me if this 'quiescent brain state' was measured in quiet sleep, active sleep or both. Why waking not examined? It seems a natural contrast to what is reported in sleep.

I have serious concerns about the methodology employed from surgery to recordings:

a) 30 minutes of post-surgical recovery (from invasive and painful procedures) is very short. Days of recovery following similar procedures are typically used in rodents, including in neonatal rodents. For example, Rensing et al., 2018 (using neonatal mice) allowed 1 hour immediate post-operative recovery, followed by at least 1 full day of recovery in the home cage with the dam, before electrophysiological recordings began. I don't find the brief recording of heart-rate during the post-surgical period sufficient to assuage these concerns.

b) I could not find specific information on the topical and systemic pain medications provided. how do they know that lingering effects of hypothermia or gas anesthesia and analgesia did not affect brain activity when recordings were made? Pain management drugs (e.g. opiate-based analgesia) can have long lasting effects in rodents-much longer than 30 minutes. Even topical lidocaine analgesia (if used) can cause lingering problems if it infiltrates tissue, especially in neonatal animals. Hence the need for hours of post-operative recovery prior to recording.

c) Were the pups acclimated to head-restraint before recordings? This does not appear to the case. Head restraint can disrupt sleep and sleeping brain activity in unacclimated rodents.

d)How do they know that the sequelae of post-surgical recovery in terms of brain activity was the same under these two different anaesthetic regimens?

e) no core temperature or ambient temperature data are provided during recordings (there is a cartoon indicating that ambient temperature was monitored, but no actual temperature data are presented for each developmental age). Neonatal rodents rapidly become hypothermic (which effects behavior and brain activity) when removed from the brood or dam; this is aggravated even further when anesthesia and obviously ice-based hypothermia is used.

Given these concerns, I find the neonatal mouse data very hard to interpret. Some of these concerns could be ameliorated by allowing longer recovery (at least 1-2 hours) in some of the pups, and then comparing the results to the short recovery data.

The fact that IACUC approved these techniques is irrelevant to the above points.

2. One would have hoped to see data from single cell recordings from the same cortical areas where LFPs were recorded (upper cortical layers). Instead, unit recordings were made from deep cortical layers. Given differences in development of these cortical layers – including intracortical inhibition – how can the authors really compare these different types of recordings in the manner that they do? They really should measure units from upper layers as well.

3. Couldn't this 'transient quiescent' state/condition be explained as a consequence of increasing intracortical inhibition? Even an epiphenomenon of the same process? That is, that it does not necessarily have any function?

4. Much of the analyses is based on surface electrophysiological measures. These results are interesting but could be discussed in context of recent work in ferrets from the Fitzpatrick and Kaschube labs. The human data could be discussed where applicable with developing human findings from Huber and LeBourgeois.

5. There does not appear to be much specific analyses of active sleep. As stated above, its not clear if analyses were always segregated by active sleep or quiet sleep.

6. The authors should cite more than seelke et al. for studies of sleep ontogeny and brain (including EEG) organization. The basic observations in seelke et al., were reported decades before, and some of their other findings are contested. see Davis et al., Ontogeny of sleep and circadian rhythms. In: Regulation of Sleep and Circadian Rhythms. Volume 133, edn. Edited by Zee PC, Turek FW. New York: Marcel Dekker, Inc.; 1999: 19-80. and also Frank and Heller, 2003

*Reviewer #3 (Recommendations for the authors):*

1. Rewrite text and reorganize figures to increase clarity and organization of results. For example, paragraph 2 of the results shift confusingly between humans and mice. Figure 1 is unclear from which species results are taken. In Figure 3 the distributions overlap and no explanation of color is given in figure (preferred) or even in the legend. Throughout there are a number of inserts that show data for one animal or condition but these are not very helpful and it is not clear what they are trying to convey. Where possible (eg. Figure 4A) age groups should be placed in the figure.

2. Rewrite the discussion to include more concrete situations with previous findings (either rejecting a lack of quiescence outside of somatosensory cortex, or hypothesizing why S1 is unique, for example).

---

## [Author Response]

Essential revisions:The following list of essential revisions were developed after rather extensive discussion between the reviewers.1) Demonstrate more thoroughly that the pausing phenomenon is not an artifact by:a) quantify twitching and sleep state distributions across ages to show that the 'quiescence' is not a simple function of less movement or active sleep.

Thank you for identifying potential confounders for our observations. To address these, we quantified twitching rate in each animal and examined whether there were any systematic changes across age groups (Figure 2—figure supplement 4B). There was no significant difference in twitching rates across age groups (ANOVA p = 0.0861), though a weak trend toward decrease in twitching over time (P5 to P14) was found, in agreement with other studies of twitches in neonatal rodents (1-3). The lack of statistically significant change in twitch rate across groups, and the lack of nadir in twitching during our identified transition period argues against our results being a function of less twitching. This data is presented in Figure 2—figure supplement 4B with relevant statistical testing.

We furthermore analyzed the proportion of time spent in active/quiet sleep across this developmental period. As known from the literature, the most mature animals had less active sleep than the most immature animals (3-5). Although the exact amount of quiet sleep in early development remains unclear, our results fits the increasing trend of quiet sleep reported and described by other groups (4). This data is presented in Figure 2—figure supplement 4A with relevant statistical testing. Post-hoc testing did not reveal a significant difference in active sleep proportion between P5-7 and P8-9 animals, or between P8-9 and P1012 animals, indicating lack of an abrupt change in sleep proportions during the transition period that could explain our results (note that the omnibus ANOVA was significant, with post-hoc differences between the P5-7 and P10-12 and P13-14 groups). Furthermore, we specifically analyzed data from periods of immobility lasting 10 seconds or more to facilitate analysis of comparable states given the difficulty in precise scoring of active and quiet sleep in neonatal rodents (5-6). Therefore, any potential effects related to sleep state are minimized.

There was no sharp transition (or statistically significant group difference) in either feature that could account for the unique electrophysiologic features exhibited by the animals at the beginning of the second postnatal week. It would also be difficult to explain the differences in oscillation spatial extent, inter-spike interval, phase locking, and cross-frequency coupling that we observe during this time as a function of twitching or sleep state. Taken together, these data do not support the notion that the pausing phenomenon is an artifact of twitching or sleep state distributions across ages.

b) quantify the spiking from superficial layers (2-4 or 2-3) since spiking data from layers, 4-6 are used, but these layers are both very diverse in their behavior, and the least likely to be strongly correlated with spindle-bursts (maximal in layer 2-4).

We quantified spiking activity from superficial cortical layers to address this point. We used the grouping of layers 2-3 and 4-6 for this purpose to maximize integration with the data obtained using surface arrays, which capture activity primarily from the superficial cortical layers. We were also mindful of the precision of the histological methods used, and thus did not separate into more than 2 groups. We found that the superficial cortical layers followed a similar pattern to the deeper layers in regard to the spiking measures analyzed (firing rate, inter-spike interval, recruitment into spindle-band oscillations). The results of these analyses are presented, with complete quantification, in Figure 3—figure supplements 1–2 and Figure 5—figure supplement 1, and referenced in the Results text. A nadir at the beginning of the second postnatal week was demonstrated in each analysis.

c) Report all the surgical details as requested by Reviewer #2 in the Recommendations for the Authors. Present an analysis of recovery time for neural activity (ECOG or spike rate) showing that differences in surgical recovery between ages cannot account for the quiescent period.

We have reported all the surgical details requested by Reviewer #2 in the Methods section. They are listed here for ease of reference as well:

a) Recovery period – We agree with the reviewer that the recovery period should be optimized for each procedure. The high-density electrophysiological implant we perform is not amenable to chronic implantation in neonatal mice, and therefore next day recording is not possible. With this necessity to perform acute recordings, we aimed for a balance between recovery from anesthesia and prolongation of the recording period, which could increase the possibility of post-surgical complications. We minimize the duration of the surgical procedure, and also the amount of isoflurane used. Pups were allowed to recover for a **minimum** of 30 minutes (see Methods); recovery criteria based on plateau of heart rate and onset of sleep wake cycling were used on an individualized basis to determine start time for analysis of neurophysiologic data. By these criteria, all pups used for analysis had recovered by 60 minutes postoperatively. We further monitor the heart rate throughout the recovery and recording period to ensure stability. As suggested, we analyzed neurophysiological metrics (spindle band oscillation occurrence and neural spiking rate) over the recovery and recording periods to investigate whether differences in surgical recovery could account for observed effects. We found that both of these metrics had plateaued by the time data was flagged for use in further analysis (Figure 1—figure supplement 2A and C), similar to isoflurane recovery noted by others (7). Furthermore, we compared these metrics from an early section of the analyzed data to a late section of the analyzed data to investigate for stability. There was no significant difference in the amount of change in these metrics (as determined by ANOVA) between P5-7, P8-9, and P10-14 pups (Figure 2—figure supplement 2B and D, p = 0.8 for spindle band oscillations, p = 0.2 for neural spiking). Therefore, recovery from anesthesia is highly unlikely to contribute to the results presented in this work.

b) Analgesics – We used subdermal bupivacaine prior to scalp incision, systemic ketoprofen injection, and topical lidocaine to intact tissue for analgesia. No opiate-based agents were selected, due to their higher potential for effects on brain function compared to ketoprofen (8). It is not possible to rule out some effects on brain activity of these agents; however, improper pain control can disrupt brain activity itself and is unacceptable during ethical animal experimentation (9, 10). To minimize any possible confounders associated with use of minimum appropriate analgesia, the same protocol was used for animals of each age with a weight-based dosing regimen. These agents are also recommended in the literature for neonatal rodent recordings (11).

c) Head restraint – We did not acclimatize the pups to head restraint because of the potential to cause further stress to the pup in advance to the recording, and experience that habituation procedures are not necessary in infant subjects. Our head restraint allows for adjustment of head position for pup comfort (± 5mm), and is sized to each individual pup. We provide olfactory cues from the nest and “swaddling” (12) to simulate the nest environment. With this preparation, the pup enters into a normal sleep wake cycle in our experience, and others’ (11). Pups that are head fixed sleep for a similar amount of time compared to unrestrained pups, and they rarely exhibit signs of distress (such as audible vocalizations or struggling).

d) Anesthesia regimes – We used hypothermia as an anesthesia induction in P5-6 pups due to the relative ineffectiveness of isoflurane to induce anesthesia without significant cardiorespiratory depression when used alone in this age. Pups of all ages were under isoflurane maintenance anesthesia during the surgical procedure, minimizing overall differences in the anesthetic regime. Furthermore, P7 pups were induced with isoflurane (no hypothermia), and systematic differences were not observed between these pups and the P5-6 pups. Therefore, we do not ascribe subsequent electrophysiological differences to the anesthesia regime used. Inhalational anesthetics are also the shortest acting generalized anesthetic agents (7).

e) Temperature control – In initial pilot experiments, we determined that an external ambient temperature of 37 degrees Celsius was sufficient to maintain the core body temperature of pups within normal limits across ages. Our recording incubator uses heaters with feedback control to maintain this constant temperature throughout the duration of recording. As such, temperature was eliminated as a variable that could contribute to changes in electrophysiological patterns.

2) Textual changes to present a more balanced view of the unknown function of the "pause" and to temper or eliminate claims of causality that are unwarranted.

We carefully amended all parts of the manuscript to ensure unwarranted claims of causality are not present. We have balanced the discussion of potential functions of the “pause.” Please see highlighted text in Abstract, Introduction, and Discussion sections.

Reviewer #1 (Recommendations for the authors):There are places where the authors presume causality without evidence. For example, although twitches are clearly accompanied by changes in LFP power, it is not clear whether the changes in LFP power cause the twitches or vice versa or whether both are caused by some additional unobserved phenomenon. Similarly, Figure 4 demonstrates developmental changes in cross-frequency coupling. While it is reasonable to hypothesize that this is due to changes in the regularity and temporal precision of synaptic activity, this has not been tested, and other changes, such as changes in the intrinsic properties of neurons could also contribute, or could even make a greater contribution.

Thank you for pointing out these considerations. In regard to twitch-related changes in LFP power, we reasoned from previous work where the peripheral sensory system was directly modified (13, 14) and other literature on twitch-related electrophysiological activity (15). Certainly, we do not engage in perturbations to test these relationships here, so we have modified the discussion to eliminate any claim of causality.

We similarly do not test what cellular or network mechanisms are responsible for emergence of cross-frequency coupling, only that this network property emerges at a specific timepoint. Cross-frequency coupling facilitates establishment of temporal epochs during which synaptic activity may have greater effect on the recipient neurons (16, 17). We have modified the discussion of cross-frequency coupling to acknowledge that different neural mechanisms could contribute to its expression.

The discussion speculates that the pause, or distinct transition period itself plays a functionally important role, but little evidence is put forward that this is more likely than that the pause is an epiphenomenon of shifting properties of thalamocortical circuits. The balance of the discussion could be improved by acknowledging this possibility. This might enhance the impact by interesting others in experiments to test the functional role by perturbing it.

We have now highlighted the importance of attempts to perturb this transition period and investigate for its absence/modification in models of abnormal neurodevelopment. The exact function of the transition period in the absence of perturbations indeed remains speculative. We have amended the discussion to fully acknowledge this point. However, identifying a key epoch during which all of these properties we investigate shift simultaneously is notable in that it suggests a critical epoch for network maturation that could be targeted as a biomarker or intervention point.

Reviewer #2 (Recommendations for the authors):1. It is not clear to me if this 'quiescent brain state' was measured in quiet sleep, active sleep or both. Why waking not examined? It seems a natural contrast to what is reported in sleep.

We have further clarified the statements in the Methods and Results sections that describe how data was selected for analysis. We aimed to target mostly quiet/NREM sleep because sleep spindles in the mature animals are restricted to this state. We agree that examination of active/REM sleep and wakefulness could certainly be interesting complements for future work.

I have serious concerns about the methodology employed from surgery to recordings:a) 30 minutes of post-surgical recovery (from invasive and painful procedures) is very short. Days of recovery following similar procedures are typically used in rodents, including in neonatal rodents. For example, Rensing et al., 2018 (using neonatal mice) allowed 1 hour immediate post-operative recovery, followed by at least 1 full day of recovery in the home cage with the dam, before electrophysiological recordings began. I don't find the brief recording of heart-rate during the post-surgical period sufficient to assuage these concerns.

Recovery period – We agree with the reviewer that the recovery period should be optimized for each procedure. The high-density electrophysiological implant we perform is not amenable to chronic implantation, and therefore next day recording is not possible. With this necessity to perform acute recordings, we aimed for a balance between recovery from anesthesia and prolongation of the recording period, which could increase the possibility of post-surgical complications. We minimize the duration of the surgical procedure, and also the amount of isoflurane used. Pups were allowed to recover for a minimum of 30 minutes (see Methods); recovery criteria based on plateau of heart rate and onset of sleep wake cycling were used on an individualized basis to determine start time for analysis of neurophysiologic data. By these criteria, all pups used for analysis had recovered by 60 minutes post-operatively. We further monitor the heart rate throughout the recovery and recording period to ensure stability. As suggested, we analyzed neurophysiological metrics (spindle band oscillation occurrence and neural spiking rate) over the recovery and recording periods to investigate whether differences in surgical recovery could account for observed effects. We found that both of these metrics had plateaued by the time data was flagged for use in further analysis (Figure 1—figure supplement 2A and C), similar to isoflurane recovery noted by others (7). Furthermore, we calculated the change in these metrics from the beginning to the end sections the period used for analysis. There was no significant difference in the amount of change in these metrics (as determined by ANOVA) between P5-7, P8-9, and P10-14 pups (Figure 2—figure supplement 2B and D, p = 0.8 for spindle band oscillations, p = 0.2 for neural spiking). Therefore, recovery from anesthesia is highly unlikely to contribute to the results presented in this work.

b) I could not find specific information on the topical and systemic pain medications provided. how do they know that lingering effects of hypothermia or gas anesthesia and analgesia did not affect brain activity when recordings were made? Pain management drugs (e.g. opiate-based analgesia) can have long lasting effects in rodents-much longer than 30 minutes. Even topical lidocaine analgesia (if used) can cause lingering problems if it infiltrates tissue, especially in neonatal animals. Hence the need for hours of post-operative recovery prior to recording.

Analgesics – We used subdermal bupivacaine prior to scalp incision, systemic ketoprofen injection, and topical lidocaine to intact tissue for analgesia. No opiate-based agents were selected, due to their higher potential for effects on brain function compared to ketoprofen (8). It is not possible to rule out some effects on brain activity of these agents; however, improper pain control can disrupt brain activity itself and is unacceptable during ethical animal experimentation (9,10). To minimize any possible confounders associated with use of minimum appropriate analgesia, the same protocol was used for animals of each age with a weight-based dosing regimen. These agents are also recommended in the literature for neonatal rodent recordings (11).

c) Were the pups acclimated to head-restraint before recordings? This does not appear to the case. Head restraint can disrupt sleep and sleeping brain activity in unacclimated rodents.

Head restraint – We did not acclimatize the pups to head restraint because of the potential to cause further stress to the pup in advance to the recording, and experience that habituation procedures are not necessary in infant subjects. Our head restraint allows for some adjustment of head position for pup comfort, and is sized to each individual pup. We provide olfactory cues from the nest and “swaddling” (12) to simulate the nest environment. With this preparation, the pup enters into a normal sleep wake cycle in our experience, and others’ (11). Pups that are head fixed sleep as much or more than unrestrained pups, and they rarely exhibit signs of distress (such as struggling or audible vocalizations).

d)How do they know that the sequelae of post-surgical recovery in terms of brain activity was the same under these two different anaesthetic regimens?

Anesthesia regimes – We used hypothermia as an anesthesia induction in P5-6 pups due to the relative ineffectiveness of isoflurane to induce anesthesia without significant cardiorespiratory depression when used alone in this age. Pups of all ages were under isoflurane maintenance anesthesia during the surgical procedure, minimizing overall differences in the anesthetic regime. Furthermore, P7 pups were induced with isoflurane (no hypothermia), and no differences were observed between these pups and the P5-6 pups. Therefore, we do not ascribe subsequent electrophysiological differences to the anesthesia regime used. Inhalational anesthetics are also the shortest acting generalized anesthetic agents (7).

e) no core temperature or ambient temperature data are provided during recordings (there is a cartoon indicating that ambient temperature was monitored, but no actual temperature data are presented for each developmental age). Neonatal rodents rapidly become hypothermic (which effects behavior and brain activity) when removed from the brood or dam; this is aggravated even further when anesthesia and obviously ice-based hypothermia is used.

Temperature control – In initial pilot experiments, we determined that an external ambient temperature of 37 degrees Celsius was sufficient to maintain the core body temperature of pups within normal limits across ages. Our recording box uses heaters with feedback control to maintain this constant temperature throughout the duration of recording. As such, temperature was eliminated as a variable that could contribute to changes in electrophysiological patterns.

Given these concerns, I find the neonatal mouse data very hard to interpret. Some of these concerns could be ameliorated by allowing longer recovery (at least 1-2 hours) in some of the pups, and then comparing the results to the short recovery data.The fact that IACUC approved these techniques is irrelevant to the above points.

To ameliorate these concerns, we analyzed neurophysiological metrics (spindle band oscillation occurrence and neural spiking rate) over the recovery and recording periods. We found that both of these metrics had plateaued by the time data was flagged for use in further analysis (Figure 1—figure supplement 2A and C). Furthermore, we compared these metrics from an early section of the analyzed data to a late section of the analyzed data to investigate for stability. There was no significant difference in the amount of change in these metrics (as determined by ANOVA) between P5-7, P8-9, and P10-14 pups (Figure 1—figure supplement 2B and D, p = 0.8 for spindle band oscillations, p = 0.2 for neural spiking). Therefore, recovery from anesthesia is highly unlikely to cloud interpretation of data in this work.

2. One would have hoped to see data from single cell recordings from the same cortical areas where LFPs were recorded (upper cortical layers). Instead, unit recordings were made from deep cortical layers. Given differences in development of these cortical layers--including intracortical inhibition--, how can the authors really compare these different types of recordings in the manner that they do? They really should measure units from upper layers as well.

We quantified spiking activity across cortical layers to address this point. We used the grouping of layers 2-3 and 4-6 for this purpose to maximize integration with the data obtained using surface arrays, which capture activity primarily from the superficial cortical layers. We were also mindful of the precision of the histological methods used, and thus did not separate into more than 2 groups. We found that the superficial cortical layers followed a similar pattern to the deeper layers in regard to the spiking measures analyzed (firing rate, inter-spike interval, recruitment into spindle-band oscillations). The results of these analyses are presented, with complete quantification, in Figure 3—figure supplements 1–2 and Figure 5—figure supplement 1, and referenced in the Results text. A nadir at the beginning of the second postnatal week was demonstrated in each analysis.

3. Couldn't this 'transient quiescent' state/condition be explained as a consequence of increasing intracortical inhibition? Even an epiphenomenon of the same process? That is, that it does not necessarily have any function?

The cellular and molecular features that contribute to expression of the transient quiescent state certainly merit ongoing investigation. That we observe such a state in mice and humans suggests that it is a biomarker for physiologic cortical network maturation. The notion of an epiphenomenon in this context, in our opinion, is not directly applicable. On a conceptual level, epiphenomenon would be interpreted to mean that no additional information about the system is provided by its existence. Electrophysiological features are governed by channel conductance, phosphorylation state of proteins, etc, but these are not usually referred to as an epiphenomenon of fundamental cellular properties, and emergent properties of neural networks are increasingly being identified. We would argue that the fact that network properties change swiftly and simultaneously during a quiescent state provides key clues about the ways by which neural networks can shift their properties. On a practical level, an identifiable marker of developmental maturation, such as a quiescent state, allows matching of cortical development timelines across species, and in instances of putative cortical pathology (e.g. genetic mutations that affect cortical development). Therefore, we posit that identification and characterization of this state are functionally useful, regardless of whether a specific function is ascribed to the state.

4. Much of the analyses is based on surface electrophysiological measures. These results are interesting but could be discussed in context of recent work in ferrets from the Fitzpatrick and Kaschube labs. The human data could be discussed where applicable with developing human findings from Huber and LeBourgeois.

Thank you for identifying additional related work, which we have now included in the discussion and referenced.

5. There does not appear to be much specific analyses of active sleep. As stated above, it’s not clear if analyses were always segregated by active sleep or quiet sleep.

As mentioned above, we targeted predominantly quiet sleep as the focus of this work, though clear differentiation between active and quiet sleep in the most immature animals is difficult. Specific analysis of active sleep and REM sleep could provide interesting avenues of future research.

6. The authors should cite more than seelke et al. for studies of sleep ontogeny and brain (including EEG) organization. The basic observations in seelke et al., were reported decades before, and some of their other findings are contested. see Davis et al., Ontogeny of sleep and circadian rhythms. In: Regulation of Sleep and Circadian Rhythms. Volume 133, edn. Edited by Zee PC, Turek FW. New York: Marcel Dekker, Inc.; 1999: 19-80. and also Frank and Heller, 2003

We appreciate these additional references and have improved our citations to reflect this work.

Reviewer #3 (Recommendations for the authors):1. Rewrite text and reorganize figures to increase clarity and organization of results. For example, paragraph 2 of the results shift confusingly between humans and mice. Figure 1 is unclear from which species results are taken. In Figure 3 the distributions overlap and no explanation of color is given in figure (preferred) or even in the legend. Throughout there are a number of inserts that show data for one animal or condition but these are not very helpful and it is not clear what they are trying to convey. Where possible (eg. Figure 4A) age groups should be placed in the figure.

We have clarified the text for discussion of Figure 1 to ensure it is always apparent what species is referred to. We do think that it is important to place the connection between rodent and human electrophysiological patterns at the beginning of the manuscript to frame the question that we are investigating.

We have added color explanations in the figure; thank you for catching this oversight. We have notated age groups in the figure by color, and have adjusted the colors to be consistent across the figures for clarity.

When we present data that requires multiple analytic steps of quantification, we endeavor to place an example of the first analytic step to aid in understanding what aspect has been used for summary data. These have been clarified in the figure legends.

2. Rewrite the discussion to include more concrete situations with previous findings (either rejecting a lack of quiescence outside of somatosensory cortex, or hypothesizing why S1 is unique, for example).

We have included specific findings from different regions and ages in the literature in the discussion to place our results in proper context.

References:

1. Karlsson KA, Blumberg MS. The union of the state: myoclonic twitching is coupled with nuchal muscle atonia in infant rats. Behav Neurosci. 2002 Oct;116(5):912-7. doi: 10.1037//07357044.116.5.912. PMID: 12369810.

2. Seelke AM, Karlsson KA, Gall AJ, Blumberg MS. Extraocular muscle activity, rapid eye movements and the development of active and quiet sleep. Eur J Neurosci. 2005 Aug;22(4):91120. doi: 10.1111/j.1460-9568.2005.04322.x. PMID: 16115214; PMCID: PMC2672593.

3. Seelke AM, Blumberg MS. The microstructure of active and quiet sleep as cortical δ activity emerges in infant rats. Sleep. 2008 May;31(5):691-9. doi: 10.1093/sleep/31.5.691. PMID: 18517038; PMCID: PMC2398759.

4. Blumberg, M. S., and Seelke, A. M. H. (2010). The form and function of infant sleep: From muscle to neocortex. In M. S. Blumberg, J. H. Freeman, and S. R. Robinson (Eds.), Oxford library of neuroscience. Oxford handbook of developmental behavioral neuroscience (p. 391–423). Oxford University Press

5. Cirelli C, Tononi G. Cortical development, electroencephalogram rhythms, and the sleep/wake cycle. Biol Psychiatry. 2015 Jun 15;77(12):1071-8. doi: 10.1016/j.biopsych.2014.12.017. Epub 2014 Dec 24. PMID: 25680672; PMCID: PMC4444390.

6. Frank MG, Heller HC. Development of REM and slow wave sleep in the rat. Am J Physiol. 1997 Jun;272(6 Pt 2):R1792-9. doi: 10.1152/ajpregu.1997.272.6.R1792. PMID: 9227592.

7. Yang W, Chini M, Pöpplau JA, Formozov A, Dieter A, Piechocinski P, Rais C, Morellini F, Sporns O, Hanganu-Opatz IL, Wiegert JS. Anesthetics fragment hippocampal network activity, alter spine dynamics, and affect memory consolidation. PLoS Biol. 2021 Apr 1;19(4):e3001146. doi: 10.1371/journal.pbio.3001146. PMID: 33793545; PMCID: PMC8016109.

8. Mast TG, Griff ER. The effects of analgesic supplements on neural activity in the main olfactory bulb of the mouse. Comp Med. 2007 Apr;57(2):167-74. PMID: 17536617.

9. Foley PL, Kendall LV, Turner PV. Clinical Management of Pain in Rodents. Comp Med. 2019 Dec 1;69(6):468-489. doi: 10.30802/AALAS-CM-19-000048. Epub 2019 Dec 10. PMID: 31822323; PMCID: PMC6935704.

10. Hestehave S, Munro G, Christensen R, Brønnum Pedersen T, Arvastson L, Hougaard P, Abelson KSP. Is there a reasonable excuse for not providing post-operative analgesia when using animal models of peripheral neuropathic pain for research purposes? PLoS One. 2017 Nov 22;12(11):e0188113. doi: 10.1371/journal.pone.0188113. PMID: 29166664; PMCID: PMC5699849.

11. Blumberg MS, Sokoloff G, Tiriac A, Del Rio-Bermudez C. A valuable and promising method for recording brain activity in behaving newborn rodents. Dev Psychobiol. 2015 May;57(4):506-17.

doi: 10.1002/dev.21305. Epub 2015 Apr 11. PMID: 25864710; PMCID: PMC4605431.

12. Corner MA, Kwee P. Cyclic EEG and motility patterns during sleep in restrained infant rats. Electroencephalogr Clin Neurophysiol. 1976 Jul;41(1):64-72. doi: 10.1016/0013-4694(76)902157. PMID: 58769.

13. Khazipov R, Sirota A, Leinekugel X, Holmes GL, Ben-Ari Y, Buzsáki G. Early motor activity drives spindle bursts in the developing somatosensory cortex. Nature. 2004 Dec 9;432(7018):75861. doi: 10.1038/nature03132. PMID: 15592414.

14. Minlebaev M, Colonnese M, Tsintsadze T, Sirota A, Khazipov R. Early γ oscillations synchronize developing thalamus and cortex. Science. 2011 Oct 14;334(6053):226-9. doi:

10.1126/science.1210574. PMID: 21998388

15. Dooley JC, Glanz RM, Sokoloff G, Blumberg MS. Self-Generated Whisker Movements Drive

State-Dependent Sensory Input to Developing Barrel Cortex. Curr Biol. 2020 Jun

22;30(12):2404-2410.e4. doi: 10.1016/j.cub.2020.04.045. Epub 2020 May 14. PMID: 32413304; PMCID: PMC7314650.

16. Schomburg EW, Fernández-Ruiz A, Mizuseki K, Berényi A, Anastassiou CA, Koch C, Buzsáki G. Theta phase segregation of input-specific γ patterns in entorhinal-hippocampal networks. Neuron. 2014 Oct 22;84(2):470-85. doi: 10.1016/j.neuron.2014.08.051. Epub 2014 Sep 25. PMID: 25263753; PMCID: PMC4253689.

17. Buschman TJ, Miller EK. Top-down versus bottom-up control of attention in the prefrontal and posterior parietal cortices. Science. 2007 Mar 30;315(5820):1860-2. doi:

10.1126/science.1138071. PMID: 17395832.

18. Bitzenhofer SH, Sieben K, Siebert KD, Spehr M, Hanganu-Opatz IL. Oscillatory activity in developing prefrontal networks results from theta-γ-modulated synaptic inputs. Cell Rep. 2015 Apr 21;11(3):486-97. doi: 10.1016/j.celrep.2015.03.031. Epub 2015 Apr 9. PMID: 25865885.

19. Bitzenhofer SH, Pöpplau JA, Hanganu-Opatz I. Γ activity accelerates during prefrontal development. *ELife*. 2020 Nov 18;9:e56795. doi: 10.7554/*eLife*.56795. PMID: 33206597; PMCID: PMC7673781.

20. Bitzenhofer SH, Pöpplau JA, Chini M, Marquardt A, Hanganu-Opatz IL. A transient developmental increase in prefrontal activity alters network maturation and causes cognitive dysfunction in adult mice. Neuron. 2021 Apr 21;109(8):1350-1364.e6. doi:

10.1016/j.neuron.2021.02.011. Epub 2021 Mar 5. PMID: 33675685; PMCID: PMC8063718.

21. Shen J, Colonnese MT. Development of Activity in the Mouse Visual Cortex. J Neurosci. 2016

Nov 30;36(48):12259-12275. doi: 10.1523/JNEUROSCI.1903-16.2016. PMID: 27903733;

PMCID: PMC5148222

22. Colonnese MT, Shen J, Murata Y. Uncorrelated Neural Firing in Mouse Visual Cortex during Spontaneous Retinal Waves. Front Cell Neurosci. 2017 Sep 20;11:289. doi:

10.3389/fncel.2017.00289. PMID: 28979189; PMCID: PMC5611364.

23. Ellingson RJ, Peters JF. Development of EEG and daytime sleep patterns in low risk premature infants during the first year of life: longitudinal observations. Electroencephalogr Clin

Neurophysiol. 1980 Oct;50(1-2):165-71. doi: 10.1016/0013-4694(80)90333-8. PMID: 6159184.